# On Second Order Behaviour
# in Augmented Neural ODEs:
# A Short Summary

**Alexander Norcliffe**\*, **Cristian Bodnar**\*, **Ben Day**\*, **Nikola Simidjievski, Pietro Liò**
Department of Computer Science and Technology
University of Cambridge
{alin2, cb2015, bjd39, ns779, pl219}@cam.ac.uk

## Abstract

In Norcliffe et al. [13], we discussed and systematically analysed how Neural ODEs (NODEs) can learn higher-order order dynamics. In particular, we focused on second-order dynamic behaviour and analysed Augmented NODEs (ANODEs), showing that they can learn second-order dynamics with only a few augmented dimensions, but are unable to correctly model the velocity (first derivative). In response, we proposed Second Order NODEs (SONODEs), that build on top of ANODEs, but explicitly take into account the second-order physics-based inductive biases. These biases, besides making them more efficient and noise-robust when modelling second-order dynamics, make them more interpretable than ANODEs, therefore more suitable in many real-world scientific modelling applications.

## 1 Introduction

From a "simple" swinging pendulum, to complex environmental/climate processes, to celestial-body motion – dynamical systems are fundamental means of studying complex phenomena in nature and society. Ordinary Differential Equations (ODEs) are a common formalism for describing dynamical behaviour, which express the rate of change of systems' states (typically) through time. Neural Ordinary Differential Equations (NODEs) [2] refer to continuous-depth neural networks that can represent a structure of an ODE, and as such allow for learning and modelling unknown dynamics from (partially-)observed data, which often cannot be described analytically.

Neural ODEs (NODEs) can be seen as a continuous variant of ResNet models [7], whose hidden state evolves continuously according to a differential equation

$$\dot{\mathbf{x}} = f^{(v)}(\mathbf{x}, t, \theta_f), \qquad \mathbf{x}(t_0) = \mathbf{X}_0, \tag{1}$$

whose velocity is described by a neural network $f^{(v)}$ with parameters $\theta_f$ and initial position given by the points of a dataset $\mathbf{X}_0$. As shown by Chen et al. [2], the gradients can be computed through an abstract adjoint state $\mathbf{r}(t)$, once its dynamics are known. Since the introduction of NODEs, there have been many studies which address different challenges and classes of modelling problems [4, 8, 11, 15, 16, 19, 22].

Our work in Norcliffe et al. [13] presented a clear in-depth study on second-order behaviour in Neural ODEs, that can be used for modelling many dynamical systems governed by second-order laws, such as oscillators, Newton's equations etc. In particular, we introduced Second Order Neural ODEs (SONODEs), an instance of Augmented Neural ODEs (ANODEs) [4] that explicitly takes into account physics-based inductive biases. We studied the optimisation of SONODEs by generalising

---

\*corresponding authors

Workshop Paper at The Symbiosis of Deep Learning and Differential Equations Workshop at NeurIPS 2021

the adjoint sensitivity method to second-order models. We showed that some useful properties of ANODEs also extend to SONODEs, which can also lead to simpler and more intuitive solutions to several benchmark problems. Finally, through a series of experiments on synthetic and real-world dynamical systems, often with known analytic solutions, we illustrated the benefits of SONODEs that relate to efficiency, noise-robustness and interpretability when modelling second-order dynamics. This paper highlights and briefly discusses these findings. The code is available online at https://github.com/a-norcliffe/sonode.

## 2 Second Order Neural Ordinary Differential Equations

Second Order Neural ODEs (SONODEs) [11, 22], with an initial position $\mathbf{x}(t_0)$, initial velocity $\dot{\mathbf{x}}(t_0)$, and acceleration $\ddot{\mathbf{x}}$ are given by

$$\mathbf{x}(t_0) = \mathbf{X}_0, \qquad \dot{\mathbf{x}}(t_0) = g(\mathbf{x}(t_0), \theta_g), \qquad \ddot{\mathbf{x}} = f^{(a)}(\mathbf{x}, \dot{\mathbf{x}}, t, \theta_f), \qquad (2)$$

where $f^{(a)}$ is a neural network with parameters $\theta_f$. Alternatively, SONODEs can be seen as a system of coupled first-order neural ODEs with state $\mathbf{z}(t) = [\mathbf{x}(t), \mathbf{a}(t)]$:

$$\mathbf{z} = \begin{bmatrix} \mathbf{x} \\ \mathbf{a} \end{bmatrix}, \quad \dot{\mathbf{z}} = f^{(v)}(\mathbf{z}, t, \theta_f) = \begin{bmatrix} \mathbf{a} \\ f^{(a)}(\mathbf{x}, \mathbf{a}, t, \theta_f) \end{bmatrix}, \quad \mathbf{z}(t_0) = \begin{bmatrix} \mathbf{X}_0 \\ g(\mathbf{X}_0, \theta_g) \end{bmatrix}. \qquad (3)$$

This formulation shows that SONODEs are a type of ANODE [4], which append states $\mathbf{a}(t)$ to the ODE:

$$\mathbf{z} = \begin{bmatrix} \mathbf{x} \\ \mathbf{a} \end{bmatrix}, \quad \dot{\mathbf{z}} = f^{(v)}(\mathbf{z}, t, \theta_f), \quad \mathbf{z}(t_0) = \begin{bmatrix} \mathbf{X}_0 \\ g(\mathbf{X}_0, \theta_g) \end{bmatrix}, \qquad (4)$$

and as such, ANODEs inherit many useful properties, such as greater function representation than NODEs and being able to learn second order dynamics with fewer dimensions than SONODEs (see Section 4 in [13]). In contrast to ANODEs, SONODEs constrain the structure of $f^{(v)}$, and offer a way to adapt NODE's first-order adjoint method for training [2] to a second order adjoint method.

**Proposition 2.1.** *The adjoint state $\mathbf{r}(t)$ of SONODEs follows the second-order ODE*

$$\ddot{\mathbf{r}} = \mathbf{r}^T \frac{\partial f^{(a)}}{\partial \mathbf{x}} - \dot{\mathbf{r}}^T \frac{\partial f^{(a)}}{\partial \dot{\mathbf{x}}} - \mathbf{r}^T \frac{d}{dt} \left( \frac{\partial f^{(a)}}{\partial \dot{\mathbf{x}}} \right) \qquad (5)$$

The proof and boundary conditions are given in Appendix B. Since the dynamics of the abstract adjoint vector are known, its state at all times $t$ can be used to train the parameters $\theta_f$ using the integral

$$\frac{dL}{d\theta_f} = - \int_{t_n}^{t_0} \mathbf{r}^T \frac{\partial f^{(a)}}{\partial \theta_f} dt, \qquad (6)$$

where $L$ denotes the loss function and $t_n$ is the timestamp of interest. The gradient with respect to the parameters of the initial velocity network, $\theta_g$, can be found in Appendix B. To further evaluate how the second-order adjoint sensitivity method compares with the first-order adjoint-based optimisation, we compare this gradient to the one obtained through the adjoint of the first-order coupled ODE from Equation (3).

**Proposition 2.2.** *The gradient of $\theta_f$ computed through the adjoint of the coupled ODE from (3) and the gradient from (6) are equivalent. However, the latter requires at least as many matrix multiplications as the former.*

This result motivates the use of the first-order coupled ODE, as it presents computational advantages. The proof in Appendix B shows that this is due to the dynamics of the adjoint from the coupled ODE, which contain entangled representations of the adjoint. This is in contrast to the disentangled representation in Equation (5), where the adjoint state and velocity are separated. It is the entangled representation, a typical phenomenon in ANODEs, that permits fast computation of the gradients for the coupled ODE. The entangled representation also introduces further properties when ANODEs are used as a modelling choice for second-order dynamical systems as described below.

Even though SONODEs build on ANODEs, they learn second-order dynamics in different ways (see Section 5.1. in [13] or Appendix E). ANODEs learn an abstract function $F$ that at $t_0$ is equal to the

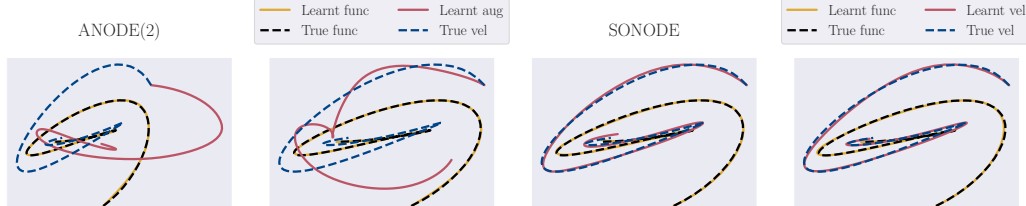

Figure 1: ANODEs and SONODEs successfully learn the trajectory in real space of a 2D ODE for two different random initialisations. However, the augmented trajectories of ANODE are in both cases widely different from the true velocity of the ODE *(left)*. In contrast, SONODE converges in both cases to the true ODE *(right)*.

initial velocity, and another function $G$ that couples to $F$ giving it the right acceleration. ANODEs learn an entangled representation of the dynamics in this way, allowing them to model dynamics with fewer dimensions (see Section 4 in [13]). In contrast, SONODEs are constrained to learn the acceleration and initial velocity directly. This leads to several useful properties, for instance when investigating the dynamics of physical systems. In the case of SONODEs, modelling this behaviour is straightforward, directly learning the acceleration as a function of position, velocity and time. However, in the case of ANODEs, this is learnt through an abstract alternative ODE where the state and augmented dimensions are entangled, impairing their interpretability.

To illustrate this, in Figure 1 we show the trajectories of an ANODE(2)[2] and a SONODE, trained to model a two-dimensional second-order system (see Equation (13) in [13]), using the correct initial velocity. Even though ANODE(2) is able to learn the true trajectory in real space, the augmented trajectories change each run and are substantially different from the true velocity of the underlying ODE. In contrast, SONODE learns the correct velocity for both runs. This clearly shows that ANODEs might not be a suitable investigative tool for scientific applications, where the physical interpretability of the results is important.

SONODEs also leverage their inductive biases to converge faster than the other models. Figure 2a shows the training loss of NODE, ANODE(1) and SONODE on a task of modelling a damped harmonic oscillator[3]. While NODEs and ANODEs learnt a general $\dot{\mathbf{z}}$, SONODEs were given $\dot{\mathbf{z}} = [v, f^{(a)}]$ and only learnt $f^{(a)}$, therefore converging in fewer iterations. Moreover, SONODEs are more robust to noise. Figure 2b illustrates the ability of ANODE(1) and SONODE to learn a sine curve ($x = \sin(t)$) in varying noise regimes. SONODEs extrapolate better than ANODEs, because they are "forced" to learn second-order dynamics, and therefore are less likely to overfit to the training data.

---

[2]ANODE(D) denotes the use of $D$ augmented dimensions.
[3]$\ddot{x} = -(\omega^2 + \gamma^2)x - 2\gamma\dot{x}$, $\gamma = 0.1$ and $\omega = 1$

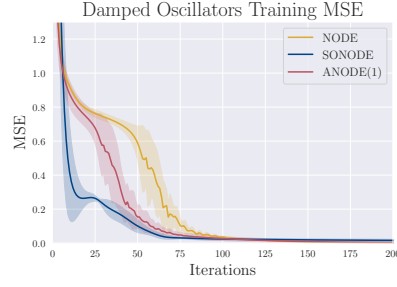

(a) NODE, ANODE(1) and SONODE training error on harmonic oscillators. SONODEs, with built-in second-order behaviour in the architectural choice, converge faster.

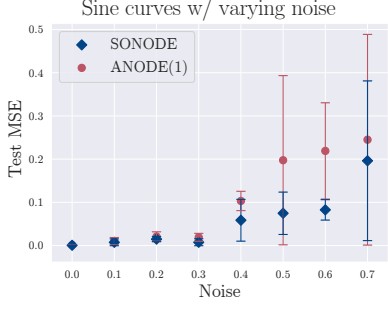

(b) SONODEs are more stable than ANODEs when learning a sine curve in different noise regimes.

Figure 2: Synthetic experiments

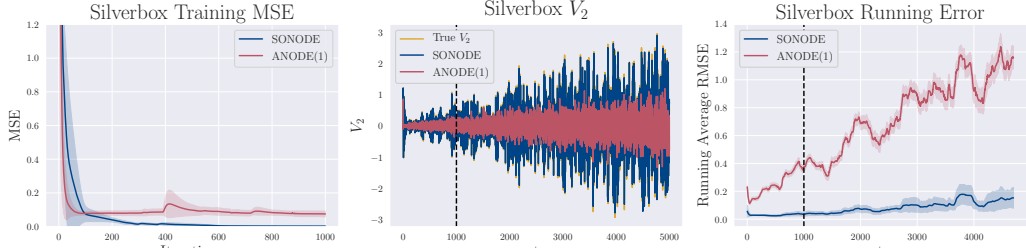

Figure 3: ANODE(1) and SONODE on on a real-world second-order Silverbox data set: training loss curves *(left)*, predicted value *(middle)*, and running error *(right)*. The models were trained on the first 1000 timestamps and extrapolated to the next 4000.

Finally, we compare SONODEs and ANODEs on a real-world modeling task. In particular, we consider modelling the Silverbox system [18], an electronic circuit resembling a Duffing Oscillator, with input voltage $V_1(t)$ and measured output $V_2(t)$. The non-linear model Silverbox represents is $\ddot{V}_2 = a\dot{V}_2 + bV_2 + cV_2^3 + dV_1$. To account for this, all models included a $V_2^3$ term. On this second-order system, shown in Figure 3, SONODEs extrapolate better than ANODEs and are able to capture the increase in the amplitude of the signal exceptionally well. Other comparisons on real-world datasets are given in Appendix D.

## 3 Discussion

SONODEs are a special case of ANODEs, whose phase-space dynamics are restricted to model second-order behaviour. We believe that for tasks where the trajectory is unimportant, and performance depends only on the endpoints (such as classification), ANODEs might perform better because they are unconstrained in how they use their capacity (see Appendix D.5). In contrast, we expect SONODEs to outperform ANODEs both in terms of accuracy and convergence rate on time-series data whose underlying dynamics is assumed (or known) to be second-order. In this setting, SONODEs have a unique functional solution and fewer local minima compared to ANODEs. At the same time, they have higher parameter efficiency since $\dot{\mathbf{x}} = \mathbf{v}$ requires no parameters, so all parameters are in the acceleration. Finally, we expect SONODEs to be more appropriate for applications in the natural sciences, where second-order dynamics are common and it is useful to recover the force equation.

Concurrent to our work [13], SONODEs have also been briefly evaluated on MNIST by Massaroli et al. [11] as part of a wider study on Neural ODEs. In contrast, our study is focused on the theoretical understanding of second-order behaviour. At the same time, our investigations are largely based on learning the dynamics of physical systems rather than classification tasks. Second-order models have also been considered in Graph Differential Equations [15] and ODE$^2$VAE [22]. In the same way SONODEs assert Newtonian mechanics, other models have been made to use physical laws, guaranteeing physically plausible results, in discrete and continuous cases. Lutter et al. [10] apply Lagrangian mechanics to cyber-physical systems, while Greydanus et al. [6] and Zhong et al. [20] use Hamiltonian mechanics to learn dynamical data.

More recently, further priors have been used on top of second order, to improve performance. Modular Neural ODEs [21] split up the acceleration into a sum of physically meaningful terms, $\ddot{\mathbf{x}} = -\nabla\phi(\mathbf{x}) + \mathbf{v} \times B(\mathbf{x}) - D(\mathbf{v})\mathbf{v}$, making up a scalar potential, magnetic field and drag force. This separation was seen to greatly improve performance, however it requires the form of the force to be known. More generally, one can use Privileged Information [17] to improve the learning of dynamics. LUPI Neural ODE Processes (learning using privileged information) [3], extend the Neural ODE Process model (NDP) [14], so that during training higher level knowledge is supplied to the model, such as conserved quantities. Then at test time the NDP is able to infer the privileged information from observations only.

## 4 Conclusion

In our work we showed that Second Order Neural ODEs (SONODEs) are a special case of Augmented Neural ODEs (ANODEs) with restricted dynamics. We showed that ANODEs can learn second

order dynamics using fewer dimensions than SONODEs, making them more efficient. However, their dimensions are entangled, making their solutions less interpretable than those learnt by SONODEs. Finally, we demonstrated with synthetic and real-world datasets, that when the dynamics are known to be second order, SONODEs are both more robust and more accurate than ANODEs.

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

## A  Phase Space Trajectory Proofs

Here we consider a $k$-th order initial value problem and its phase space properties.

**Lemma A.1.** *For a k-th order IVP, where the k-th derivative is Lipschitz continuous, a solution cannot have discontinuities in the time derivative of its phase space trajectory.*

*Proof.* Consider the phase space trajectory $\mathbf{z}(t) = \left[ \mathbf{x}(t), \dfrac{d\mathbf{x}}{dt}(t), ..., \dfrac{d^{k-1}\mathbf{x}}{dt^{k-1}}(t) \right]$. Let $f$ be the k-th time derivative of $\mathbf{x}(t)$. Then the time derivative of $\mathbf{z}(t)$ is

$$\frac{d}{dt} \begin{bmatrix} \mathbf{x} \\ \dfrac{d\mathbf{x}}{dt} \\ ... \\ \dfrac{d^{k-1}\mathbf{x}}{dt^{k-1}} \end{bmatrix} = \begin{bmatrix} \dfrac{d\mathbf{x}}{dt} \\ \dfrac{d^2\mathbf{x}}{dt^2} \\ ... \\ f(\mathbf{z}) \end{bmatrix}$$

If for one set of finite arguments, $\mathbf{z}_1$, $f(\mathbf{z_1})$ is also finite, then because the gradients of $f$ are all bounded (due to Lipschitz continuity), for any other finite arguments, $\mathbf{z}_n$, $f(\mathbf{z}_n)$ will remain finite. Now consider $\dfrac{d^{k-1}\mathbf{x}}{dt^{k-1}}$, its time derivative is $f(\mathbf{z}(t))$, which is finite for all finite $\mathbf{z}$. Therefore, $\dfrac{d^{k-1}\mathbf{x}}{dt^{k-1}}$, can't have discontinuities with a finite derivative, and also must be finite for finite $\mathbf{z}$. Now consider $\dfrac{d^{k-2}\mathbf{x}}{dt^{k-2}}$, its time derivative is finite for all finite $\mathbf{z}$, and therefore it can't have discontinuities and also must be finite for all finite $\mathbf{z}$. This line of argument continues up to $\mathbf{x}$. The state $\mathbf{x}$ and all of its time derivatives up to the $k$-th have no discontinuities and are finite. Therefore as long as the initial conditions $\mathbf{z}(t_0)$ are finite, there can be no discontinuities in the time derivative of the phase space trajectory at finite time. □

**Proposition A.2.** *For a k-th order IVP, if the k-th derivative of $\mathbf{x}$ is Lipschitz continuous and has no explicit time dependence, then unique phase space trajectories cannot intersect at an angle. Similarly, a single phase space trajectory cannot intersect itself at an angle.*

*Proof.* Consider two trajectories $\mathbf{z}_1(t)$ and $\mathbf{z}_2(t)$ that have different initial conditions $\mathbf{z}_1(t_0) = \mathbf{h}_1$ and $\mathbf{z}_2(t_0) = \mathbf{h}_2$. Assume the trajectories cross at a point in phase space at an angle, $\mathbf{z}_1(t_1) = \mathbf{z}_2(t_2) = \tilde{\mathbf{h}}$. If they intersect at an angle, then evolving the two states by a small time $\delta t \ll 1$, and using the Lipschitz continuity of $f$, meaning that the trajectories cannot have kinks in them (as shown in Lemma A.1), $\mathbf{z}_1(t_1 + \delta t) \neq \mathbf{z}_2(t_2 + \delta t)$. However, if they are at the same point in phase space, then they must have the same k-th order derivative, $f$. All other derivatives are equal, so by evolving the states by the same small time $\delta t \ll 1$, $\mathbf{z}_1(t_1 + \delta t) = \mathbf{z}_2(t_2 + \delta t)$. There is a contradiction and therefore the assumption is wrong, unique trajectories cannot cross at an angle in phase space when $f$ is Lipschitz continuous and has no $t$ dependence.

Now consider the single trajectory $\mathbf{z}(t)$. Assume it intersects itself at an angle, at $t_1$ and $t_2$. Now consider two particles on this trajectory, starting at $t_1 - \tau$ and $t_2 - \tau$ such that $t_2 - \tau > t_1$. These two particles have different initial conditions and cross at an angle. However, the above shows that cannot happen. Therefore, the assumption that $\mathbf{z}(t)$ can intersect itself at an angle must be wrong. Trajectories cannot intersect themselves in phase space at an angle. □

Trajectories can, however, feed into each other representing the same particle path at different times. Single phase space trajectories can feed into themselves representing periodic motion. This requires a Lipschitz continuous $f$, and for there to be no explicit time dependence. If there was time dependence then two trajectories can cross at different times, and a trajectory can self intersect. Effectively an additional dimension is added to phase space, which is time. The propositions above would still hold because $\dfrac{dt}{dt} = 1$ which is Lipschitz continuous. Therefore, with time included as a phase space dimension, intersections in space are only forbidden if they occur at the same time.

# B   Adjoint Sensitivity Method

We present a proof to both the first and second order Adjoint method, using a Lagrangian style approach [1, 5]. We also prove that when the underlying ODE is second order, using the first order method on a concatenated state, $\mathbf{z} = [\mathbf{x}, \mathbf{v}]$, produces the same results as the second order method but does so more efficiently. All parameters, $\theta$, are time-independent (so $\frac{d\theta}{dt} = \frac{dt}{d\theta} = 0$).

## B.1   First Order Adjoint Method

Let L denote a scalar loss function, $L = L(\mathbf{x}(t_n))$, the gradient with respect to a parameter $\theta$ is

$$\frac{dL}{d\theta} = \frac{\partial L}{\partial \mathbf{x}(t_n)^T} \frac{d\mathbf{x}(t_n)}{d\theta} \tag{7}$$

The vector $\dfrac{\partial L}{\partial \mathbf{x}(t_n)^T}$ is found using backpropagation. For dynamical data the loss will depend on multiple time stamps, there is also a sum over timestamps, $t_n$. Therefore $\dfrac{d\mathbf{x}(t_n)}{d\theta}$ is needed. $\mathbf{x}(t_n)$ follows

$$\mathbf{x}(t_n) = \int_{t_0}^{t_n} \dot{\mathbf{x}}(t) dt + \mathbf{x}(t_0) \tag{8}$$

subject to

$$\dot{\mathbf{x}} = f^{(v)}(\mathbf{x}, t, \theta_f), \qquad \mathbf{x}(t_0) = s(\mathbf{X}_0, \theta_s) \tag{9}$$

where $\mathbf{X}_0$ is the data going into the network and is constant. The functions $f^{(v)}$ and $s$ describe the ODE and the initial conditions. Here we allow $\mathbf{X}_0$ to first go through the transformation, $s(\mathbf{X}_0, \theta_s)$. This maintains generality and allows NODEs to be used as a component of a larger model. For example, $\mathbf{X}_0$ could go through a ResNet before the NODE, and then through a softmax classifier at the end (which is accounted for in the term $\dfrac{\partial L}{\partial \mathbf{x}(t_n)^T}$). Introduce the new variable $\mathbf{F}$

$$\mathbf{F} = \int_{t_0}^{t_n} \dot{\mathbf{x}}(t) dt = \int_{t_0}^{t_n} \left( \dot{\mathbf{x}} + A(t)(\dot{\mathbf{x}} - f^{(v)}) \right) dt + B(\mathbf{x}(t_0) - s) \tag{10}$$

These are equivalent because $(\dot{\mathbf{x}} - f^{(v)})$ and $(\mathbf{x}(t_0) - s)$ are both zero. This means the matrices, $A(t)$ and $B$, can be chosen freely (as long as they are well behaved, finite etc.), to make the computation easier. The gradients of $\mathbf{x}(t_n)$ with respect to the parameters are

$$\frac{d\mathbf{x}(t_n)}{d\theta_f} = \frac{d\mathbf{F}}{d\theta_f}, \qquad \frac{d\mathbf{x}(t_n)}{d\theta_s} = \frac{d\mathbf{F}}{d\theta_s} + \frac{ds(\mathbf{X}_0, \theta_s)}{d\theta_s} \tag{11}$$

Differentiating $\mathbf{F}$ with respect to a general parameter $\theta$

$$\frac{d\mathbf{F}}{d\theta} = \int_{t_0}^{t_n} \frac{d\dot{\mathbf{x}}}{d\theta} dt + \int_{t_0}^{tn} A(t) \left( \frac{d\dot{\mathbf{x}}}{d\theta} - \frac{\partial f^{(v)}}{\partial \theta} - \frac{\partial f^{(v)}}{\partial \mathbf{x}^T} \frac{d\mathbf{x}}{d\theta} \right) dt + B \left( \frac{d\mathbf{x}(t_0)}{d\theta} - \frac{ds}{d\theta} \right) \tag{12}$$

Integrating by parts

$$\int_{t_0}^{t_n} A(t) \frac{d\dot{\mathbf{x}}}{d\theta} dt = \left[ A(t) \frac{d\mathbf{x}}{d\theta} \right]_{t_0}^{t_n} - \int_{t_0}^{t_n} \dot{A}(t) \frac{d\mathbf{x}}{d\theta} dt \tag{13}$$

Substituting this in and using $\int_{t_0}^{t_n} \frac{d\dot{\mathbf{x}}}{d\theta} dt = [\frac{d\mathbf{x}}{d\theta}]_{t_0}^{t_n}$, gives

$$\begin{aligned}
\frac{d\mathbf{F}}{d\theta} = & \left( \frac{d\mathbf{x}}{d\theta} + A(t)\frac{d\mathbf{x}}{d\theta} \right)\bigg|_{t_n} - \left( \frac{d\mathbf{x}}{d\theta} + A(t)\frac{d\mathbf{x}}{d\theta} \right)\bigg|_{t_0} - \int_{t_0}^{t_n} A(t) \frac{\partial f^{(v)}}{\partial \theta} dt \\
& - \int_{t_0}^{t_n} \left( \dot{A}(t) + A(t)\frac{\partial f^{(v)}}{\partial \mathbf{x}^T} \right) \frac{d\mathbf{x}}{d\theta} dt + B \left( \frac{d\mathbf{x}}{d\theta}\bigg|_{t_0} - \frac{ds}{d\theta} \right)
\end{aligned} \tag{14}$$

Using the freedom of choice of $A(t)$, let it follow the ODE

$$\dot{A}(t) = -A(t)\frac{\partial f^{(v)}}{\partial \mathbf{x}^T}, \qquad A(t_n) = -I \tag{15}$$

Where $I$ is the identity matrix. Then the first term and second integral in Equation (14) become zero, yielding

$$\frac{d\mathbf{F}}{d\theta} = (B - I - A(t_0))\left.\frac{d\mathbf{x}}{d\theta}\right|_{t_0} + \int_{t_n}^{t_0} A(t)\frac{\partial f^{(v)}}{\partial \theta}dt - B\frac{ds}{d\theta} \tag{16}$$

Now using the freedom of choice of $B$, let it obey the equation

$$B = I + A(t_0) \tag{17}$$

This makes the first term in Equation (16) zero. This gives the final form of $\dfrac{d\mathbf{F}}{d\theta}$

$$\frac{d\mathbf{F}}{d\theta} = \int_{t_n}^{t_0} A(t)\frac{\partial f^{(v)}}{\partial \theta}dt - (I + A(t_0))\frac{ds}{d\theta} \tag{18}$$

Subbing into Equation (11) and using the fact that $f^{(v)}$ has no $\theta_s$ dependence and $s$ has no $\theta_f$ dependence

$$\frac{d\mathbf{x}(t_n)}{d\theta_f} = \int_{t_n}^{t_0} A(t)\frac{\partial f^{(v)}(\mathbf{x}, t, \theta_f)}{\partial \theta_f}dt, \qquad \frac{d\mathbf{x}(t_n)}{d\theta_s} = -A(t_0)\frac{ds(\mathbf{X}_0, \theta_s)}{d\theta_s} \tag{19}$$

This leads to the gradients of the loss

$$\frac{dL}{d\theta_f} = \frac{\partial L}{\partial \mathbf{x}(t_n)^T}\int_{t_n}^{t_0} A(t)\frac{\partial f^{(v)}(\mathbf{x}, t, \theta_f)}{\partial \theta_f}dt, \qquad \frac{dL}{d\theta_s} = -\frac{\partial L}{\partial \mathbf{x}(t_n)^T}A(t_0)\frac{ds(\mathbf{X}_0, \theta_s)}{d\theta_s} \tag{20}$$

Subject to the ODE for $A(t)$

$$\dot{A}(t) = -A(t)\frac{\partial f^{(v)}(\mathbf{x}, t, \theta_f)}{\partial \mathbf{x}}, \qquad A(t_n) = -I \tag{21}$$

Now introduce the adjoint state $\mathbf{r}(t)$

$$\mathbf{r}(t) = -A(t)^T\frac{\partial L}{\partial \mathbf{x}(t_n)}, \qquad \mathbf{r}(t)^T = -\frac{\partial L}{\partial \mathbf{x}(t_n)^T}A(t) \tag{22}$$

Using the fact that $\dfrac{\partial L}{\partial \mathbf{x}(t_n)}$ is constant with respect to time, the adjoint equations are obtained by applying the definition of the adjoint in Equation (22), to the gradients in Equation (20), and multiplying the ODE in Equation (21) by the constant $-\frac{\partial L}{\partial \mathbf{x}(t_n)}$

$$\frac{dL}{d\theta_f} = -\int_{t_n}^{t_0} \mathbf{r}(t)^T\frac{\partial f^{(v)}(\mathbf{x}, t, \theta_f)}{\partial \theta_f}dt, \qquad \frac{dL}{d\theta_s} = \mathbf{r}(t_0)^T\frac{ds(\mathbf{X}_0, \theta_s)}{d\theta_s} \tag{23}$$

Where the adjoint $\mathbf{a}(t)$ follows the ODE

$$\dot{\mathbf{r}}(t) = -\mathbf{r}(t)^T\frac{\partial f^{(v)}(\mathbf{x}, t, \theta_f)}{\partial \mathbf{x}}, \qquad \mathbf{r}(t_n) = \frac{\partial L}{\partial \mathbf{x}(t_n)} \tag{24}$$

The gradients are found by integrating the adjoint state, $\mathbf{r}$, and the real state, $\mathbf{x}$, backwards in time, which requires no intermediate values to be stored, using constant memory, a major benefit over traditional backpropagation.

These are the same equations that were derived by Chen et al. [2], however this includes the addition of letting $\mathbf{x}(t_0) = s(\mathbf{X}_0, \theta_s)$ giving the corresponding gradient, $\dfrac{dL}{d\theta_s}$. Additionally, the derivation used by Chen et al. [2] is simpler but does not present an obvious way to extend the adjoint method to second order ODEs, which this derivation method can do, as shown next.

## B.2 Second Order Adjoint

Using the same derivation method, but with a second order differential equation, a second order adjoint method is derived, according to the proposition from the main text:

**Proposition 2.1.** *The adjoint state* $\mathbf{r}(t)$ *of SONODEs follows the second order ODE*

$$\ddot{\mathbf{r}} = \mathbf{r}^T \frac{\partial f^{(a)}}{\partial \mathbf{x}} - \dot{\mathbf{r}}^T \frac{\partial f^{(a)}}{\partial \dot{\mathbf{x}}} - \mathbf{r}^T \frac{d}{dt} \left( \frac{\partial f^{(a)}}{\partial \dot{\mathbf{x}}} \right) \tag{25}$$

*and the gradients of the loss with respect to the parameters of the acceleration,* $\theta_f$ *are*

$$\frac{dL}{d\theta_f} = -\int_{t_n}^{t_0} \mathbf{r}^T \frac{\partial f^{(a)}}{\partial \theta_f} dt, \tag{26}$$

*Proof.* In general, the loss function, $L$, depends on $\mathbf{x}$ and $\dot{\mathbf{x}}$

$$\frac{dL}{d\theta} = \frac{\partial L}{\partial \mathbf{x}(t_n)^T} \frac{d\mathbf{x}(t_n)}{d\theta} + \frac{\partial L}{\partial \dot{\mathbf{x}}(t_n)^T} \frac{d\dot{\mathbf{x}}(t_n)}{d\theta} \tag{27}$$

The gradients from the positional part and the velocity part are found separately and added. Firstly the position

$$\mathbf{x}(t_n) = \int_{t_0}^{t_n} \dot{\mathbf{x}}(t) dt + \mathbf{x}(t_0) \tag{28}$$

Subject to the second order ODE

$$\ddot{\mathbf{x}} = f^{(a)}(\mathbf{x}, \dot{\mathbf{x}}, t, \theta_f), \qquad \mathbf{x}(t_0) = s(\mathbf{X}_0, \theta_s), \qquad \dot{\mathbf{x}}(t_0) = g(\mathbf{x}(t_0), \theta_g) \tag{29}$$

Following the same procedure as in first order, but including the initial condition for the velocity as well

$$\mathbf{F} = \int_{t_0}^{t_n} \dot{\mathbf{x}} + A(t)(\ddot{\mathbf{x}} - f^{(a)})dt + B(\dot{\mathbf{x}}(t_0) - g) + C(\mathbf{x}(t_0) - s) \tag{30}$$

As before, the vectors, $(\ddot{\mathbf{x}} - f^{(a)})$, $(\dot{\mathbf{x}}(t_0) - g)$ and $(\mathbf{x}(t_0) - s)$ are zero, which gives freedom to choose the matrices $A(t)$, $B$ and $C$ to make the calculation easier. The gradients of $\mathbf{x}(t_n)$ with respect to the parameters $\theta$ are

$$\frac{d\mathbf{x}(t_n)}{d\theta_f} = \frac{d\mathbf{F}}{d\theta_f}, \qquad \frac{d\mathbf{x}(t_n)}{d\theta_g} = \frac{d\mathbf{F}}{d\theta_g}, \qquad \frac{d\mathbf{x}(t_n)}{d\theta_s} = \frac{d\mathbf{F}}{d\theta_s} + \frac{ds(\mathbf{X}_0, \theta_s)}{d\theta_s} \tag{31}$$

Differentiating $F$ from equation 30 with respect to a general parameter

$$\begin{aligned}
\frac{d\mathbf{F}}{d\theta} &= \left[ \frac{d\mathbf{x}}{d\theta} \right]_{t_0}^{t_n} - \int_{t_0}^{t_n} A(t) \frac{\partial f^{(a)}}{\partial \theta} dt + \int_{t_0}^{t_n} A(t) \left( \frac{d\ddot{\mathbf{x}}}{d\theta} - \frac{\partial f^{(a)}}{\partial \mathbf{x}^T} \frac{d\mathbf{x}}{d\theta} - \frac{\partial f^{(a)}}{\partial \dot{\mathbf{x}}^T} \frac{d\dot{\mathbf{x}}}{d\theta} \right) dt \\
&+ B \left( \frac{d\dot{\mathbf{x}}}{d\theta} \bigg|_{t_0} - \frac{\partial g}{\partial \theta} - \frac{\partial g}{\partial \mathbf{x}(t_0)^T} \frac{d\mathbf{x}(t_0)}{d\theta} \right) + C \left( \frac{d\mathbf{x}}{d\theta} \bigg|_{t_0} - \frac{ds}{d\theta} \right)
\end{aligned} \tag{32}$$

Integrating by parts

$$\int_{t_0}^{t_n} A(t) \frac{d\ddot{\mathbf{x}}}{d\theta} dt = \left[ A(t) \frac{d\dot{\mathbf{x}}}{d\theta} - \dot{A}(t) \frac{d\mathbf{x}}{d\theta} \right]_{t_0}^{t_n} + \int_{t_0}^{t_n} \ddot{A}(t) \frac{d\mathbf{x}}{d\theta} dt \tag{33}$$

$$\int_{t_0}^{t_n} A(t) \frac{\partial f^{(a)}}{\partial \dot{\mathbf{x}}^T} \frac{d\dot{\mathbf{x}}}{d\theta} dt = \left[ A(t) \frac{\partial f^{(a)}}{\partial \dot{\mathbf{x}}^T} \frac{d\mathbf{x}}{d\theta} \right]_{t_0}^{t_n} - \int_{t_0}^{t_n} \frac{d}{dt} \left( A(t) \frac{\partial f^{(a)}}{\partial \dot{\mathbf{x}}^T} \right) \frac{d\mathbf{x}}{d\theta} dt \tag{34}$$

Subbing these into Equation (32)

$$\frac{d\mathbf{F}}{d\theta} = \left[\left(I - \dot{A} - A\frac{\partial f^{(a)}}{\partial \dot{\mathbf{x}}^T}\right)\frac{d\mathbf{x}}{d\theta} + A\frac{d\dot{\mathbf{x}}}{d\theta}\right]_{t_n} - \left[\left(I - \dot{A} - A\frac{\partial f^{(a)}}{\partial \dot{\mathbf{x}}}\right)\frac{d\mathbf{x}}{d\theta} + A\frac{d\dot{\mathbf{x}}}{d\theta}\right]_{t_0}$$
$$+ \int_{t_0}^{t_n}\left(\ddot{A}(t) - A(t)\frac{\partial f^{(a)}}{\partial \mathbf{x}^T} + \frac{d}{dt}\left(A(t)\frac{\partial f^{(a)}}{\partial \dot{\mathbf{x}}^T}\right)\right)\frac{d\mathbf{x}}{d\theta}dt + \int_{t_n}^{t_0} A(t)\frac{\partial f^{(a)}}{\partial \theta}dt \qquad (35)$$
$$+ B\left(\left.\frac{d\dot{\mathbf{x}}}{d\theta}\right|_{t_0} - \frac{\partial g}{\partial \theta} - \frac{\partial g}{\partial \mathbf{x}(t_0)^T}\frac{d\mathbf{x}(t_0)}{d\theta}\right) + C\left(\left.\frac{d\mathbf{x}}{d\theta}\right|_{t_0} - \frac{ds}{d\theta}\right)$$

Using the freedom to choose $A(t)$, let it follow the second order ODE

$$\ddot{A}(t) = A(t)\frac{\partial f^{(a)}}{\partial \mathbf{x}^T} - \frac{d}{dt}\left(A(t)\frac{\partial f^{(a)}}{\partial \dot{\mathbf{x}}^T}\right), \qquad A(t_n) = 0, \qquad \dot{A}(t_n) = I \qquad (36)$$

This makes the first term and first integral in Equation (35) zero, yielding

$$\frac{d\mathbf{F}}{d\theta} = \int_{t_n}^{t_0} A(t)\frac{\partial f^{(a)}}{\partial \theta}dt + \left(\left(\dot{A}(t) + A(t)\frac{\partial f^{(a)}}{\partial \dot{\mathbf{x}}^T} - I - B\frac{\partial g}{\partial \mathbf{x}(t_0)^T} + C\right)\frac{d\mathbf{x}}{d\theta}\right)\bigg|_{t_0}$$
$$+ \left((B - A)\frac{d\dot{\mathbf{x}}}{d\theta}\right)\bigg|_{t_0} - B\frac{\partial g}{\partial \theta} - C\frac{ds}{d\theta} \qquad (37)$$

Now using the freedom of choice in $B$ and $C$

$$B = A(t_0), \qquad C = -\dot{A}(t_0) - A(t_0)\frac{\partial f^{(a)}}{\partial \dot{\mathbf{x}}}\bigg|_{t_0} + I + A(t_0)\frac{\partial g}{\partial \mathbf{x}(t_0)^T} \qquad (38)$$

This makes the second and third terms in Equation (37) zero, yielding

$$\frac{d\mathbf{F}}{d\theta} = \int_{t_n}^{t_0} A(t)\frac{\partial f^{(a)}}{\partial \theta}dt - B\frac{\partial g}{\partial \theta} - C\frac{ds}{d\theta} \qquad (39)$$

These give the final gradients of $\mathbf{x}(t_n)$ with respect to the parameters, by subbing the results for $B$, $C$ and $\frac{d\mathbf{F}}{d\theta}$ above into Equation (31), using the fact that $f^{(a)}$, $g$ and $s$ only depend on the parameters $\theta_f$, $\theta_g$ and $\theta_s$ respectively

$$\frac{d\mathbf{x}(t_n)}{d\theta_f} = \int_{t_n}^{t_0} A(t)\frac{\partial f^{(a)}}{\partial \theta_f}dt, \qquad \frac{d\mathbf{x}(t_n)}{d\theta_g} = -A(t_0)\frac{\partial g}{\partial \theta_g}$$
$$\frac{d\mathbf{x}(t_n)}{d\theta_s} = \left(\dot{A}(t_0) + A(t_0)\left(\frac{\partial f^{(a)}}{\partial \dot{\mathbf{x}}^T}\bigg|_{t_0} - \frac{\partial g}{\partial \mathbf{x}(t_0)^T}\right)\right)\frac{ds}{d\theta_s} \qquad (40)$$

As before, introduce the adjoint state $\mathbf{r}^x(t)$:

$$\mathbf{r}^x(t) = -A(t)^T\frac{\partial L}{\partial \mathbf{x}(t_n)}, \qquad \mathbf{r}^x(t)^T = -\frac{\partial L}{\partial \mathbf{x}(t_n)^T}A(t) \qquad (41)$$

Using the fact that $\frac{\partial L}{\partial \mathbf{x}(t_n)}$ is constant with respect to time, all the results above, and the ODE and initial conditions for $A(t)$ in Equation (36) can be multiplied by $-\frac{\partial L}{\partial \mathbf{x}(t_n)^T}$, to get the gradients $\frac{dL}{d\theta}$ in terms of $\mathbf{r}^x(t)$

$$\frac{dL}{d\theta_f} = -\int_{t_n}^{t_0} \mathbf{r}^x(t)^T\frac{\partial f^{(a)}}{\partial \theta_f}dt, \qquad \frac{dL}{d\theta_g} = \mathbf{r}^x(t_0)^T\frac{\partial g}{\partial \theta_g}$$
$$\frac{dL}{d\theta_s} = \left(-\dot{\mathbf{r}}^x(t_0)^T - \mathbf{r}^x(t_0)^T\left(\frac{\partial f^{(a)}}{\partial \dot{\mathbf{x}}^T}\bigg|_{t_0} - \frac{\partial g}{\partial \mathbf{x}(t_0)^T}\right)\right)\frac{d\mathbf{x}(t_0)}{d\theta_s} \qquad (42)$$

Subject to the second order ODE for $\mathbf{r}^x(t)$

$$\ddot{\mathbf{r}}^x(t) = \mathbf{r}^x(t)^T \frac{\partial f^{(a)}}{\partial \mathbf{x}} - \frac{d}{dt}\left(\mathbf{r}^x(t)^T \frac{\partial f^{(a)}}{\partial \dot{\mathbf{x}}}\right), \qquad \mathbf{r}^x(t_n) = 0, \qquad \dot{\mathbf{r}}^x(t_n) = -\frac{\partial L}{\partial \mathbf{x}(t_n)} \qquad (43)$$

Where after differentiating with the product rule the ODE in Equation (43) becomes

$$\ddot{\mathbf{r}}^x(t) = \mathbf{r}^x(t)^T \frac{\partial f^{(a)}}{\partial \mathbf{x}} - \dot{\mathbf{r}}^x(t)^T \frac{\partial f^{(a)}}{\partial \dot{\mathbf{x}}} - \mathbf{r}^x(t)^T \left(\frac{d}{dt}\frac{\partial f^{(a)}}{\partial \dot{\mathbf{x}}}\right) \qquad (44)$$

Where doing the full time derivative gives

$$\frac{d}{dt}\left(\frac{\partial f^{(a)}}{\partial \dot{\mathbf{x}}}\right) = [\dot{\mathbf{x}}^T, f^{(a)T}, 1]\begin{bmatrix}\partial_{\mathbf{x}} \\ \partial_{\dot{\mathbf{x}}} \\ \partial_t\end{bmatrix}\left(\frac{\partial f^{(a)}}{\partial \dot{\mathbf{x}}}\right) \qquad (45)$$

Where the fact that $\ddot{x} = f^{(a)}$ has been used. This is only when the loss depends on the position. The same method is used to look at the velocity part in Equation (27)

$$\frac{dL}{d\theta} = \frac{\partial L}{\partial \dot{\mathbf{x}}(t_n)^T}\frac{d\dot{\mathbf{x}}(t_n)}{d\theta} \qquad (46)$$

Where

$$\dot{\mathbf{x}}(t_n) = \int_{t_0}^{t_n} \ddot{\mathbf{x}}(t)dt + \dot{\mathbf{x}}(t_0) \qquad (47)$$

The general method is to take this expression and add zeros, in the form of $A(t)$, $B$ and $C$ multiplied by the ODE and initial conditions, $(\ddot{\mathbf{x}} - f^{(a)})$, $(\dot{\mathbf{x}}(t_0) - g)$ and $(\mathbf{x}(t_0) - s)$. Then differentiate with respect to a general parameter $\theta$ and integrate by parts to get any integrals containing $\frac{d\dot{\mathbf{x}}}{d\theta}$ or $\frac{d\ddot{\mathbf{x}}}{d\theta}$ in terms of $\frac{d\mathbf{x}}{d\theta}$. Then choose the ODE for $A(t)$ to remove any $\frac{d\mathbf{x}}{d\theta}$ terms in the integral, and the initial conditions of $A(t_n)$ to remove the boundary terms at $t_n$. Then $B$ and $C$ are chosen to remove the boundary terms at $t_0$. After doing this the gradients of $\dot{\mathbf{x}}$ with respect to the parameters are

$$\frac{d\dot{\mathbf{x}}(t_n)}{d\theta_f} = \int_{t_n}^{t_0} A(t)\frac{\partial f^{(a)}}{\partial \theta_f}dt, \qquad \frac{d\dot{\mathbf{x}}(t_n)}{d\theta_g} = -A(t_0)\frac{\partial g}{\partial \theta_g}$$

$$\frac{d\dot{\mathbf{x}}(t_n)}{d\theta_s} = \left(\dot{A}(t_0) + A(t_0)\frac{\partial f^{(a)}}{\partial \dot{\mathbf{x}}^T}\bigg|_{t_0} - A(t_0)\frac{\partial g}{\partial \mathbf{x}(t_0)^T}\right)\frac{ds}{d\theta_s} \qquad (48)$$

Subject to the second order ODE for $A(t)$

$$\ddot{A}(t) = A(t)\frac{\partial f^{(a)}}{\partial \mathbf{x}^T} - \frac{d}{dt}\left(A(t)\frac{\partial f^{(a)}}{\partial \dot{\mathbf{x}}^T}\right), \qquad A(t_n) = -I, \qquad \dot{A}(t_n) = \frac{\partial f^{(a)}}{\partial \dot{\mathbf{x}}^T}\bigg|_{t_n} \qquad (49)$$

Now introduce the state $\mathbf{r}^v(t)$

$$\mathbf{r}^v(t) = -\frac{\partial L}{\partial \dot{\mathbf{x}}(t_n)^T}A(t), \qquad \mathbf{r}^v(t) = -A(t)^T\frac{\partial L}{\partial \dot{\mathbf{x}}(t_n)} \qquad (50)$$

Which allows the gradients of the loss with respect to the parameters to be written as

$$\frac{dL}{d\theta_f} = -\int_{t_n}^{t_0} \mathbf{r}^v(t)^T \frac{\partial f^{(a)}}{\partial \theta_f}dt, \qquad \frac{dL}{d\theta_g} = \mathbf{r}^v(t_0)^T\frac{\partial g}{\partial \theta_g}$$

$$\frac{dL}{d\theta_s} = \left(\mathbf{r}^v(t_0)^T\frac{\partial g}{\partial \mathbf{x}(t_0)^T} - \dot{\mathbf{r}}^v(t_0)^T - \mathbf{r}^v(t_0)^T\frac{\partial f^{(a)}}{\partial \dot{\mathbf{x}}^T}\bigg|_{t_0}\right)\frac{ds}{d\theta_s} \qquad (51)$$

Where $\mathbf{r}^v$ follows the second order ODE and initial conditions

$$\ddot{\mathbf{r}}^v(t) = \mathbf{r}^v(t)^T \frac{\partial f^{(a)}}{\partial \mathbf{x}} - \dot{\mathbf{r}}^v(t)^T \frac{\partial f^{(a)}}{\partial \dot{\mathbf{x}}} - \mathbf{r}^v(t)^T \frac{d}{dt} \left( \frac{\partial f^{(a)}}{\partial \dot{\mathbf{x}}} \right)$$

$$\mathbf{r}^v(t_n) = \frac{\partial L}{\partial \dot{\mathbf{x}}(t_n)}, \qquad \dot{\mathbf{r}}^v(t_n) = -\frac{\partial L}{\partial \dot{\mathbf{x}}(t_n)^T} \frac{\partial f^{(a)}}{\partial \dot{\mathbf{x}}} \bigg|_{t_n}$$

(52)

Now adding the gradients from the $\mathbf{x}$ dependence and the $\dot{\mathbf{x}}$ dependence together. It can be seen that the gradients are the same in Equations (42) and (51), but just swapping $\mathbf{r}^x$ and $\mathbf{r}^v$. Additionally, it can be seen from the ODEs for $\mathbf{r}^x$ and $\mathbf{r}^v$ in Equations (44) and (52), that they are governed by the same, linear, second order ODE, with different initial conditions. Therefore the gradients, $\frac{dL}{d\theta}$, can be written in terms of a new adjoint state, $\mathbf{r} = \mathbf{r}^x + \mathbf{r}^v$

$$\frac{dL}{d\theta_f} = -\int_{t_n}^{t_0} \mathbf{r}(t)^T \frac{\partial f^{(a)}(\mathbf{x}, \dot{\mathbf{x}}, t, \theta_f)}{\partial \theta_f} dt, \qquad \frac{dL}{d\theta_g} = \mathbf{r}(t_0)^T \frac{\partial g(\mathbf{x}(t_0), \theta_g)}{\partial \theta_g}$$

$$\frac{dL}{d\theta_s} = \left( \mathbf{r}(t_0)^T \frac{\partial g(\mathbf{x}(t_0), \theta_g)}{\partial \mathbf{x}(t_0)^T} - \dot{\mathbf{r}}(t_0)^T - \mathbf{r}(t_0)^T \frac{\partial f^{(a)}(\mathbf{x}, \dot{\mathbf{x}}, t, \theta_f)}{\partial \dot{\mathbf{x}}^T} \bigg|_{t_0} \right) \frac{ds(\mathbf{X}_0, \theta_s)}{d\theta_s}$$

(53)

Where $\mathbf{a}$ follows the second order ODE with initial conditions

$$\ddot{\mathbf{r}}(t) = \mathbf{r}(t)^T \frac{\partial f^{(a)}(\mathbf{x}, \dot{\mathbf{x}}, t, \theta_f)}{\partial \mathbf{x}} - \dot{\mathbf{r}}(t) \frac{\partial f^{(a)}(\mathbf{x}, \dot{\mathbf{x}}, t, \theta_f)}{\partial \dot{\mathbf{x}}} - \mathbf{r}(t)^T \frac{d}{dt} \left( \frac{\partial f^{(a)}(\mathbf{x}, \dot{\mathbf{x}}, t, \theta_f)}{\partial \dot{\mathbf{x}}} \right)$$

$$\mathbf{r}(t_n) = \frac{\partial L}{\partial \dot{\mathbf{x}}(t_n)}, \qquad \dot{\mathbf{r}}(t_n) = -\frac{\partial L}{\partial \mathbf{x}(t_n)} - \frac{\partial L}{\partial \dot{\mathbf{x}}(t_n)^T} \frac{\partial f^{(a)}(\mathbf{x}, \dot{\mathbf{x}}, t, \theta_f)}{\partial \dot{\mathbf{x}}} \bigg|_{t_n}$$

(54)

The full derivative, $d_t(\partial_{\dot{\mathbf{x}}} f^{(a)})$, is given by Equation (45). The ODE can also be written compactly as

$$\ddot{\mathbf{r}}(t) = \mathbf{r}(t)^T \frac{\partial f^{(a)}(\mathbf{x}, \dot{\mathbf{x}}, t, \theta_f)}{\partial \mathbf{x}} - \frac{d}{dt} \left( \mathbf{r}(t)^T \frac{\partial f^{(a)}(\mathbf{x}, \dot{\mathbf{x}}, t, \theta_f)}{\partial \dot{\mathbf{x}}} \right)$$

(55)

Just as in the first order method, a sum over times stamps $t_n$ may be required. This matches and extends on the gradients and ODE given by proposition 2.1. $\qquad\square$

### B.3 Equivalence between the two Adjoint methods

When acting on a concatenated state, $\mathbf{z}(t) = [\mathbf{x}(t), \mathbf{v}(t)]$, the first order adjoint method will produce the same gradients as the second order adjoint method. However, it is more computationally efficient to use the first order method. This is also given in the main text as the following proposition:

**Proposition 2.2.** *The gradient of $\theta_f$ computed through the adjoint of the coupled ODE from (3) and the gradient from (6) are equivalent. However, the latter requires at least as many matrix multiplications as the former.*

Intuitively, the first order method will produce the same gradients because second order dynamics can be thought of as two coupled first order ODEs, where the first order dynamics happen in phase space. However, this provides no information about computational efficiency. We prove the equivalence and compare the computational efficiencies below.

*Proof.* The first order formulation of second order dynamics can be written as

$$\mathbf{z}(t) = \begin{bmatrix} \mathbf{x}(t) \\ \mathbf{v}(t) \end{bmatrix}, \qquad \dot{\mathbf{z}} = \begin{bmatrix} \mathbf{v} \\ f^{(a)}(\mathbf{x}, \mathbf{v}, t, \theta_f) \end{bmatrix}, \qquad \mathbf{z}(t_0) = \begin{bmatrix} \mathbf{x}(t_0) \\ \mathbf{v}(t_0) \end{bmatrix} = \begin{bmatrix} s(\mathbf{X}_0, \theta_s) \\ g(s(\mathbf{X}_0, \theta_s), \theta_g) \end{bmatrix}$$

(56)

When using index notation, $x_i$ and $v_i$ are concatenated to make $z_i$. For $x_i$ and $v_i$, the index, i, ranges from 1 to $d$, whereas for $z_i$ it ranges from 1 to $2d$ accounting for the concatenation. This is

represented below

$$z_i = \begin{cases} x_i, & \text{if } i \leq d \\ v_{(i-d)}, & \text{if } i \geq d+1 \end{cases} \tag{57}$$

It also extends to $\dot{z}_i$ and $z_i(t_0)$, where $f_i^{(a)}$, $s_i$ and $g_i$ also have the index range from 1 to $d$, but the index of $\dot{z}_i$ goes from 1 to $2d$ just like for $z_i$.

$$\dot{z}_i = \tilde{f}_i^{(v)}(\mathbf{z}, t, \tilde{\theta}_f) = \begin{cases} v_i, & \text{if } i \leq d \\ f_{(i-d)}^{(a)}(\mathbf{x}, \mathbf{v}, t, \theta_f), & \text{if } i \geq d+1 \end{cases} \tag{58}$$

$$z_i(t_0) = \tilde{s}_i(\mathbf{X}_0, \tilde{\theta}_s) = \begin{cases} s_i(\mathbf{X}_0, \theta_s), & \text{if } i \leq d \\ g_{(i-d)}(s(\mathbf{X}_0, \theta_s), \theta_g), & \text{if } i \geq d+1 \end{cases} \tag{59}$$

Using the first order adjoint method, Equations (23) and (24), and using index notation with repeated indices summed over, the gradients are

$$\frac{dL}{d\tilde{\theta}_f} = -\int_{tn}^{t_0} r_i(t)\frac{\partial \tilde{f}_i^{(v)}(\mathbf{z}, t, \tilde{\theta}_f)}{\partial \tilde{\theta}_f}\, dt, \qquad \frac{dL}{d\tilde{\theta}_s} = r_i(t_0)\frac{d\tilde{s}_i(\mathbf{X}_0, \tilde{\theta}_s)}{d\tilde{\theta}_s} \tag{60}$$

Where the adjoint follows the ODE

$$\dot{r}_i(t) = -r_j(t)\frac{\partial \tilde{f}_j^{(v)}(\mathbf{z}, t, \tilde{\theta}_f)}{\partial z_i}, \qquad r_i(t_n) = \frac{\partial L}{\partial z_i(t_n)} \tag{61}$$

Where just like in $z_i$, the index, i, ranges from 1 to $2d$ in the adjoint $r_i(t)$. When writing the sum over the index explicitly

$$\dot{r}_i = -\sum_{j=1}^{2d} r_j \frac{\partial \tilde{f}_j^{(v)}}{\partial z_i} \qquad = -\sum_{j=1}^{d} r_j \frac{\partial \tilde{f}_j^{(v)}}{\partial z_i} - \sum_{j=d+1}^{2d} r_j \frac{\partial \tilde{f}_j^{(v)}}{\partial z_i} \tag{62}$$

Now split up the adjoint state, $\mathbf{r}$, into two equally sized vectors, $\mathbf{r}^A$ and $\mathbf{r}^B$, where their indices only range from 1 to $d$, like $\mathbf{x}$, $\mathbf{v}$, $f^{(a)}$, $g$ and $s$.

$$r_i = \begin{cases} r_i^A, & \text{if } i \leq d \\ r_{(i-d)}^B, & \text{if } i \geq d+1 \end{cases} \tag{63}$$

Using Equations (57), (58), (59) and (63), and subbing them into Equation (62), the derivative can be written as

$$\dot{r}_i = -\sum_{j=1}^{d} r_j^A \frac{\partial v_j}{\partial z_i} - \sum_{j=d+1}^{2d} r_{(j-d)}^B \frac{\partial f_{(j-d)}^{(a)}}{\partial z_i} \tag{64}$$

Relabelling the indices in the second sum $(j-d) \rightarrow j$

$$\dot{r}_i = -\sum_{j=1}^{d} r_j^A \frac{\partial v_j}{\partial z_i} - \sum_{j=1}^{d} r_j^B \frac{\partial f_j^{(a)}}{\partial z_i} \tag{65}$$

Looking at specific values of i:

$i \leq d$

$$\dot{r}_i = \dot{r}_i^A = -\sum_{j=1}^{d} r_j^A \frac{\partial v_j}{\partial x_i} - \sum_{j=1}^{d} r_j^B \frac{\partial f_j^{(a)}}{\partial x_i}, \qquad = -\sum_{j=1}^{d} r_j^B \frac{\partial f_j^{(a)}}{\partial x_i} \tag{66}$$

$i \geq d+1$

$$\dot{r}_i = \dot{r}_{(i-d)}^B = -\sum_{j=1}^{d} r_j^A \frac{\partial v_j}{\partial v_{(i-d)}} - \sum_{j=1}^{d} r_j^B \frac{\partial f_j^{(a)}}{\partial v_{(i-d)}} \tag{67}$$

Relabelling the first index $(i - d) \to i$

$$\dot{r}_i^B = -\sum_{j=1}^{d} r_j^A \frac{\partial v_j}{\partial v_i} - \sum_{j=1}^{d} r_j^B \frac{\partial f_j^{(a)}}{\partial v_i} \tag{68}$$

Noting that, $\frac{\partial v_j}{\partial v_i} = \delta_{ij}$, the time derivatives can be written in vector matrix notation as

$$\dot{\mathbf{r}}^A(t) = -\mathbf{r}^B(t)^T \frac{\partial f^{(a)}(\mathbf{x}, \mathbf{v}, t, \theta_f)}{\partial \mathbf{x}} \tag{69}$$

$$\dot{\mathbf{r}}^B(t) = -\mathbf{r}^A(t) - \mathbf{r}^B(t)^T \frac{\partial f^{(a)}(\mathbf{x}, \mathbf{v}, t, \theta_f)}{\partial \mathbf{v}} \tag{70}$$

Differentiating Equation (70), and using Equation (69) for $\dot{\mathbf{r}}^A(t)$

$$\ddot{\mathbf{r}}^B(t) = \mathbf{r}^B(t)^T \frac{\partial f^{(a)}(\mathbf{x}, \mathbf{v}, t, \theta_f)}{\partial \mathbf{x}} - \frac{d}{dt} \left( \mathbf{r}^B(t)^T \frac{\partial f^{(a)}(\mathbf{x}, \mathbf{v}, t, \theta_f)}{\partial \mathbf{v}} \right) \tag{71}$$

This matches the ODE for the second order method in Equation (55). Now applying the initial conditions, using index notation again

$$r_i(t_n) = \frac{\partial L}{\partial z_i(t_n)} \tag{72}$$

For $i \leq d$

$$r_i = r_i^A(t_n) = \frac{\partial L}{\partial x_i(t_n)} \tag{73}$$

For $i \geq d + 1$

$$r_i(t_n) = r_{(i-d)}^B(t_n) = \frac{\partial L}{\partial v_{(i-d)}(t_n)} \quad \to \quad r_i^B(t_n) = \frac{\partial L}{\partial v_i(t_n)} \tag{74}$$

Applying these initial conditions in $\mathbf{r}^A$ and $\mathbf{r}^B$ to Equation (70)

$$\dot{r}_i^B(t_n) = -\frac{\partial L}{\partial x_i(t_n)} - \frac{\partial L}{\partial v_j(t_n)} \frac{\partial f_j^{(a)}}{\partial v_i} \bigg|_{t_n} \tag{75}$$

By looking at the ODE and initial conditions, it is clear $\mathbf{r}^B$ is equivalent to the second order adjoint, in Equation (54). Now looking at the gradients, and including an explicit sum over the index

$$\frac{dL}{d\tilde{\theta}_f} = -\int_{t_n}^{t_0} \sum_{i=1}^{2d} r_i \frac{\partial \tilde{f}_i^{(v)}}{\partial \tilde{\theta}_f} dt \quad \to \quad = -\int_{t_n}^{t_0} \sum_{i=1}^{d} r_i^A \frac{\partial v_i}{\partial \tilde{\theta}_f} dt - \int_{t_n}^{t_0} \sum_{i=d+1}^{2d} r_{(i-d)}^B \frac{\partial f_{(i-d)}^{(a)}}{\partial \tilde{\theta}_f} dt \tag{76}$$

The first term is zero because $v$ has no explicit $\theta$ dependence. The second term, after relabelling and using summation convention becomes

$$\frac{dL}{d\theta_f} = -\int_{t_n}^{t_0} r_i^B(t) \frac{\partial f_i^{(a)}}{\partial \theta_f} dt \quad = -\int_{t_n}^{t_0} \mathbf{r}^B(t)^T \frac{\partial f^{(a)}}{\partial \theta_f} dt \tag{77}$$

Where $\tilde{\theta}_f = \theta_f$ has been used, as they are both the parameters for the acceleration. This matches the result for gradients of parameters in the acceleration term $\theta_f$, when using the second order adjoint method, because $\mathbf{r}^B$ is the adjoint.

Looking at the gradients related to the initial conditions

$$\frac{dL}{d\tilde{\theta}_s} = \mathbf{r}(t_0)^T \frac{d\tilde{s}(\mathbf{X}_0, \tilde{\theta}_s)}{d\tilde{\theta}_s} \tag{78}$$

After going through the previous process of separating out the sums from $1 \to d$ and $d+1 \to 2d$, then relabelling the indices on $\mathbf{r}^B$, this becomes

$$= r_i^A(t_0) \frac{ds_i(\mathbf{X}_0, \theta_s)}{d\tilde{\theta}_s} + r_i^B(t_0) \frac{dg_i(s(\mathbf{X}_0, \theta_s), \theta_g)}{d\tilde{\theta}_s} \tag{79}$$

Using the expression for $\mathbf{r}^A$ by rearranging Equation (70), this can be written as

$$\frac{dL}{d\tilde{\theta}_s} = \left( -\dot{r}_i^B(t_0) - r_j^B(t_0) \frac{\partial f_j^{(a)}}{\partial v_i} \bigg|_{t_0} \right) \frac{ds_i}{d\tilde{\theta}_s} + r_i^B(t_0) \frac{dg_i}{d\tilde{\theta}_s} \tag{80}$$

The parameters $\tilde{\theta}_s$ contain both $\theta_s$ and $\theta_g$. Looking at $\theta_g$ first, where $s(\mathbf{X}_0, \theta_s)$ has no dependence

$$\frac{dL}{d\theta_g} = r_i^B(t_0) \frac{\partial g_i(s(\mathbf{X}_0, \theta_s), \theta_g)}{\partial \theta_g} = \mathbf{r}^B(t_0)^T \frac{\partial g(s(\mathbf{X}_0, \theta_s), \theta_g)}{\partial \theta_g} \tag{81}$$

where $\frac{dg}{d\theta_g}$ can be written as a partial derivative, because $\mathbf{X}_0$ and $\theta_s$ have no dependence on $\theta_g$.

This expression is equivalent to $\frac{dL}{d\theta_g}$ found using the second order adjoint method. Now looking at the parameters $\theta_s$, these parameters are in $s(\mathbf{X}_0, \theta_s)$ explicitly and $g(s, \theta_g)$, implicitly through $s$. Subbing $\tilde{\theta}_s = \theta_s$ into Equation (80) gives

$$\frac{dL}{d\theta_s} = \left( -\dot{r}_i^B(t_0) - r_j^B(t_0) \frac{\partial f_j^{(a)}(\mathbf{x}, \mathbf{v}, t, \theta_f)}{\partial v_i} \bigg|_{t_0} + r_j^B(t_0) \frac{\partial g_j(s(\mathbf{X}_0, \theta_s), \theta_g)}{\partial s_i} \right) \frac{ds_i(\mathbf{X}_0, \theta_s)}{d\theta_s} \tag{82}$$

Using the fact that $\mathbf{x}(t_0) = s$, this is the same result for $\frac{dL}{d\theta_s}$ found using the second order adjoint method:

$$\frac{dL}{d\theta_s} = \left( \mathbf{r}^B(t_0)^T \frac{\partial g(\mathbf{x}(t_0), \theta_g)}{\partial \mathbf{x}(t_0)^T} - \dot{\mathbf{r}}^B(t_0)^T - \mathbf{r}^B(t_0)^T \frac{\partial f^{(a)}(\mathbf{x}, \mathbf{v}, t, \theta_f)}{\partial \mathbf{v}^T} \bigg|_{t_0} \right) \frac{ds(\mathbf{X}_0, \theta_s)}{d\theta_s} \tag{83}$$

All of the gradients match, so the first order adjoint method acting on $\mathbf{z}(t) = [\mathbf{x}(t), \mathbf{v}(t)]$ will produce the same gradients as the second order adjoint method acting on $\mathbf{x}(t)$. Given by Equation (53).

Looking at the efficiencies of each method and how they would be implemented. Both methods would integrate the state $\mathbf{z} = [\mathbf{x}, \mathbf{v}]$ forward in time, with $\dot{\mathbf{z}} = [\mathbf{v}, f^{(a)}]$. Both methods then integrate $\mathbf{z}$ and the adjoint backwards, in the same way. The difference is how the adjoint is represented. In first order it is represented as $[\mathbf{r}^A, \mathbf{r}^B]$ where $\mathbf{r}^B$ is the adjoint, in second order it is represented as $[\mathbf{r}, \dot{\mathbf{r}}]$ where $\mathbf{r}$ is the adjoint.

The time derivatives and initial conditions for the first order adjoint representation are

$$\begin{aligned}
\frac{d}{dt} \mathbf{r}^A(t) &= -\mathbf{r}^B(t)^T \frac{\partial f^{(a)}(\mathbf{x}, \mathbf{v}, t, \theta_f)}{\partial \mathbf{x}} \\
\frac{d}{dt} \mathbf{r}^B(t) &= -\mathbf{r}^A(t) - \mathbf{r}^B(t)^T \frac{\partial f^{(a)}(\mathbf{x}, \mathbf{v}, t, \theta_f)}{\partial \mathbf{v}} \\
\mathbf{r}^A(t_n) &= \frac{\partial L}{\partial \mathbf{x}(t_n)} \\
\mathbf{r}^B(t_n) &= \frac{\partial L}{\partial \mathbf{v}(t_n)}
\end{aligned} \tag{84}$$

The time derivatives and intial conditions for the second order adjoint representation are

$$\frac{d}{dt}\mathbf{r}(t) = \dot{\mathbf{r}}(t)$$

$$\frac{d}{dt}\dot{\mathbf{r}}(t) = \mathbf{r}(t)^T \frac{\partial f^{(a)}(\mathbf{x}, \mathbf{v}, t, \theta_f)}{\partial \mathbf{x}} - \dot{\mathbf{r}}(t)^T \frac{\partial f^{(a)}(\mathbf{x}, \mathbf{v}, t, \theta_f)}{\partial \mathbf{v}} - \mathbf{r}(t)^T \frac{d}{dt}\left(\frac{\partial f^{(a)}(\mathbf{x}, \mathbf{v}, t, \theta_f)}{\partial \mathbf{v}}\right)$$

$$\mathbf{r}(t_n) = \frac{\partial L}{\partial \mathbf{v}(t_n)}$$

$$\dot{\mathbf{r}}(t_n) = -\frac{\partial L}{\partial \mathbf{x}(t_n)} - \frac{\partial L}{\partial \mathbf{v}(t_n)^T} \frac{\partial f^{(a)}(\mathbf{x}, \mathbf{v}, t, \theta_f)}{\partial \mathbf{v}}\bigg|_{t_n}$$

(85)

Where

$$\frac{d}{dt}\left(\frac{\partial f^{(a)}}{\partial \mathbf{v}}\right) = [\mathbf{v}^T, f^{(a)T}, 1]\begin{bmatrix}\partial_\mathbf{x} \\ \partial_\mathbf{v} \\ \partial_t\end{bmatrix}\left(\frac{\partial f^{(a)}}{\partial \mathbf{v}}\right) \tag{86}$$

Looking at Equations (84) and (85), the second order method has the additional term, $\mathbf{r} \cdot d_t(\partial_\mathbf{v}(f^{(a)}))$, in the ODE, and the additional term, $(\partial_\mathbf{v} L) \cdot (\partial_\mathbf{v} f^{(a)})$ in the initial conditions. The first order method acting on the concatenated state, $[\mathbf{x}, \mathbf{v}]$, requires equal or fewer matrix multiplications than the second order method acting on $\mathbf{x}$, to find the gradients at each step and the initial conditions. This is in the general case, but also for all specific cases, it is as efficient or more efficient. The same is also true for calculating the final gradients. □

The reason for the difference in efficiencies is the state, $\mathbf{r}^B$, is the adjoint, and the state, $\mathbf{r}^A$, contains a lot of the complex information about the adjoint. It is an entangled representation of the adjoint, contrasting with the disentangled second order representation $[\mathbf{r}, \dot{\mathbf{r}}]$. This is similar to how ANODEs can learn an entangled representation of second order ODEs and SONODEs learn the disentangled representation, seen in Section E. However, entangled representations are more useful here, because they do not need to be interpretable, they just need to produce the gradients, and the entangled representation can do this more efficiently.

This analysis provides useful information on the inner workings of the adjoint method. It shows a second order specific method does exist, but the first order method acting on a state $\mathbf{z} = [\mathbf{x}, \mathbf{v}]$ will produce the same gradients more efficiently, due to how it represents the complexity. This was specific to second order ODEs, however, the first order adjoint will work on any system of ODEs, because any motion can be thought of as being first order motion in phase space. Additionally, the first order method may be the most efficient adjoint method. The complexity going from the first order to the second order was seen based on the calculation, so this is only likely to get worse as the system of ODEs becomes more complicated.

## C    Properties of SONODEs

In this section, we analyse certain properties of SONODEs and illustrate them with toy examples.

### C.1    Generalised parity problem

It is known that unique trajectories in NODEs cannot cross at the same time [4, 11]. We extend this to higher order Initial Value Problems (IVP). Proofs are presented in Appendix A.

**Proposition C.1.** *For a k-th order IVP, if the k-th derivative of $\mathbf{x}$ is Lipschitz continuous and has no explicit time dependence, then unique phase space trajectories cannot intersect at an angle. Similarly, a single phase space trajectory cannot intersect itself at an angle.*

While this shows SONODE trajectories cannot cross in phase space, they can cross in real space if they have different velocities. To illustrate this, we introduce a generalised parity problem, an extension to $D$ dimensions of the $g_{1d}$ function from Dupont et al. [4], which maps $\mathbf{x} \rightarrow -\mathbf{x}$. We

remark that SONODEs should be able to learn a parity flip in any number of dimensions, with a trivial solution

$$f^{(a)}(\mathbf{x}, \dot{\mathbf{x}}, t, \theta_f) = 0, \qquad g(\mathbf{x}(t_0), \theta_g) = -\frac{2}{t_N - t_0}\mathbf{x}(t_0) \qquad (87)$$

This is equivalent to all points moving in straight lines through the origin to $-\mathbf{x}(t_0)$.

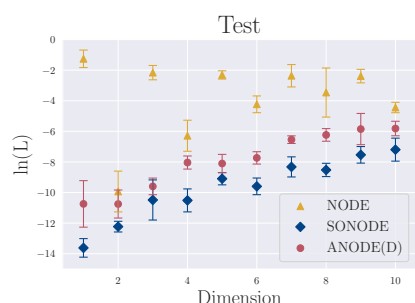

Figure 4: The logarithm of the loss in each dimension for the generalised parity problem. SONODE has the lowest loss, while the NODE loss generally oscillates between dimensions as predicted.

For higher dimensions, we first remark that NODEs are able to produce parity flips for even dimensions by pairing off the dimensions and performing a $180°$ rotation in each pair. This solution does not apply to odd-dimensional cases because there is always an unpaired dimension that is not rotated. In addition to the dimensional-parity effect, as volume increases exponentially with the dimensionality, the density exponentially decreases (given the number of points in the dataset remains constant). This makes it easier to manipulate the points without trajectories crossing, and so, it is expected that the problem will become easier for NODEs as dimensionality increases.

In Figure 4, we investigate parity flips in higher dimensions, using 50 training points and 10 test points, randomly generated between -1 and 1 in each dimension. For NODEs, as predicted, the loss oscillates over dimensions and, for odd dimensions, the loss decreases with the number of dimensions. ANODEs perform better than NODES, especially in odd dimensions, where it can rotate the unpaired dimension through the additional space. SONODEs have the lowest loss in every generalisation, which can be associated with the existence of the trivial solution in any number of dimensions, given by Equation (87).

## C.2 Nested n-spheres

Dupont et al. [4] prove that a transformation under NODEs has to be a homeomorphism, preserving the topology of the input space, and as such, they cannot learn certain transformations. Similarly to ANODEs, SONODEs avoid this problem.

**Proposition C.2.** *SONODEs are not restricted to homeomorphic transformations in real space.*

The proof can be found in Appendix C.3. To illustrate this, we perform an experiment on the nested n-spheres problem [4], (the name is taken from [11], originally called $g$ function [4]), where the elements of the blue class are surrounded by the elements of the red class (Figure 5) such that a homeomorphic transformation in that space cannot linearly separate the two classes. As expected, only ANODEs and SONODEs can learn a mapping.

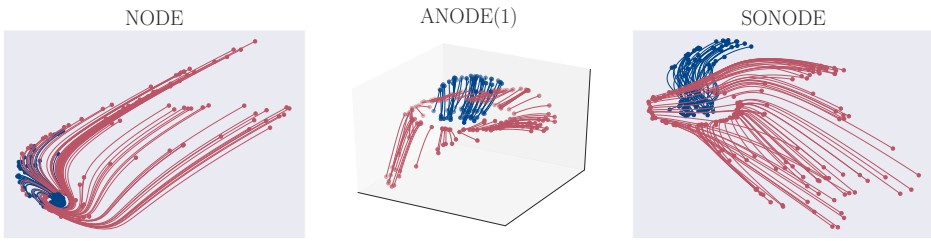

Figure 5: The trajectories learnt by NODE *(left)*, ANODE *(middle)* and SONODE *(right)* for the nested n-spheres problem in 2D. NODE preserves the topology, so the blue region cannot escape the red region. ANODE, as expected, uses the third dimension to separate the two regions. For SONODE, the points pass through each other in real space.

## C.3 Second Order ODEs are not Homeomorphisms

One of the conditions for a transformation to be a homeomorphism is for the transformation to be bijective (one-to-one and onto). In real space, a transformation that evolves according to a second order ODE does not have to be one-to-one. This is demonstrated using a one-dimensional counter-example

$$\ddot{x} = 0 \qquad \rightarrow \qquad x(t) = x_0 + v_0 t$$

$$x_0 = \begin{bmatrix} [0] \\ [1] \end{bmatrix}, \qquad v_0 = -x_0 + 2 = \begin{bmatrix} [2] \\ [1] \end{bmatrix}$$

If $t_0 = 0$ and $t_N = 1$

$$x(1) = \begin{bmatrix} [2] \\ [2] \end{bmatrix}$$

So the transformation in real space is not always one-to-one, and therefore, not always a homeomorphism.

# D   Experimental Setup and Additional Results

We anticipate two main uses for SONODEs. One is using an experiment in a controlled environment, where the aim is to find values such as the coefficient of friction. The other use is when data is observed, and the aim is to extrapolate in time, but the experiment is not controlled, for example, observing weather. We would expect for the former, a simple model with only a single linear layer would be useful, to find those coefficients, and for the latter, a deeper model may be more appropriate. Additionally, Neural ODEs may be used in classification or other tasks that only involve the start and endpoints of the flow. For all of these tasks we used $t_0 = 0$ and $t_1 = 1$, and accelerations that were not time-dependent. For tasks depending on the start and endpoint only, a deeper neural network is more useful for the acceleration.

For all experiments, except the MNIST experiment, we optimise using Adam with a learning rate of 0.01. We also train on the complete datasets and do not minibatch. All the experiments were repeated 3 times to obtain a mean and standard deviation. Depending on the task at hand, we used two different architectures for NODEs, ANODEs and SONODEs. The first is a simple linear model, one weight matrix and bias without activations. This architecture, in the case of NODEs, ANODEs and SONODEs, was used on Silverbox, Airplane Vibrations and Van-Der-Pol Oscillator, with the aim of extracting coefficients from the models, for these tasks we also allowed ANODEs to learn the initial augmented position. The second architecture is a fully connected network with two hidden layers of size 20, it uses ELU activations in $\dot{z}$ and tanh activations in the initial conditions. ELU and tanh were used because they allow for negative values in the ODE [11].

When considering ANODEs, they are in a higher-dimensional space than the problem, and the result must be projected down to the lower dimensions. This projection was not learnt as a linear layer, instead, the components were directly selected, using an identity for the real dimensions, and zero for the augmented dimensions. This was done because a final (or initial) learnt linear layer would hide the advantages of certain models. For example, the parity problem can be solved easily if NODEs are given a final linear layer, do not move the points and then multiply by -1. For this reason, no models used a linear layer at the end of the flow. Equally, they do not initialise with a linear layer as they again hide advantages. For example, the nested n-spheres problem, NODEs can solve this with an initial linear layer, if they were to go into a higher-dimensional space the points may already be linearly separated, as shown by Massaroli et al. [11].

## D.1   Airplane vibrations

The dataset [12] concerns real vibrations measurements of an airplane. A shaker was attached underneath the right wing, producing an acceleration $a_1$. Additional accelerations at different points were measured including $a_2$, which was examined in this experiment, the acceleration on the right wing, next to a non-linear interface of interest. This is a higher order system, therefore it pertains to be a challenging modelling task. The results presented in Figure 6 show that while both methods can model the dynamics reasonably well, ANODEs perform marginally better. We conjecture that this is

due to ANODEs not being restricted to second order behaviour, allowing them to partially access higher order dynamics. We test this conjecture in Appendix D.3.

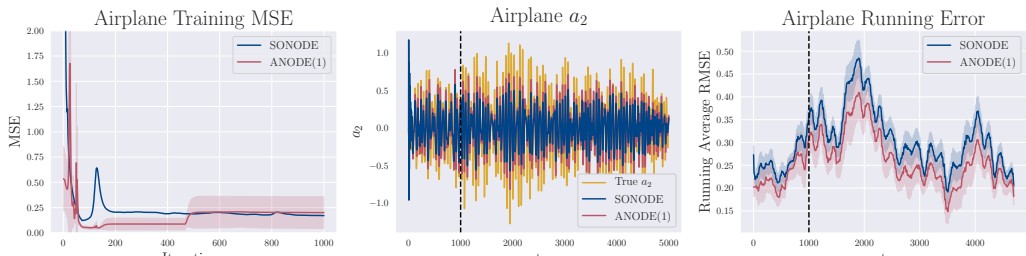

Figure 6: ANODE(1) and SONODE on the Airplane Vibrations dataset: training loss curves *(left)*, predicted value *(middle)*, and running error *(right)*. The models were trained on the first 1000 timestamps and then extrapolated to the next 4000. ANODEs are able to perform slightly better than SONODEs because they are able to access higher order dynamics.

## D.2 Van Der Pol Oscillator

ANODEs and SONODEs were tested on a forced Van Der Pol (VDP) Oscillator that exhibits chaotic behaviour. More specifically, the parameters and equations of the particular VDP oscillator are:

$$\ddot{x} = 8.53(1 - x^2)\dot{x} - x + 1.2\cos(0.2\pi t), \qquad x_0 = 0.1, \qquad \dot{x}_0 = 0 \qquad (88)$$

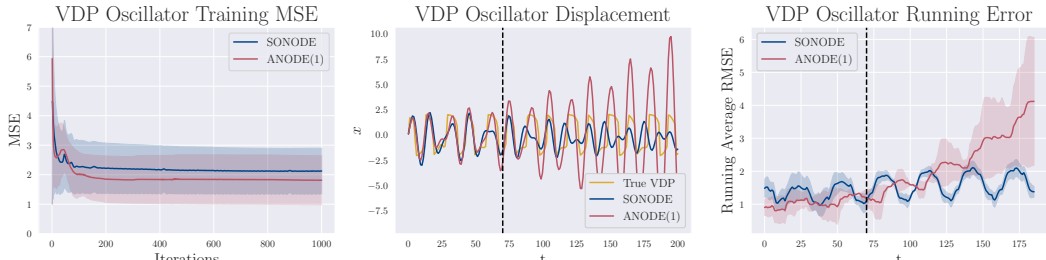

Figure 7: ANODE(1) and SONODE learning a Van-Der-Pol Oscillator: training loss curves *(left)*, predicted value *(middle)*, and running error *(right)*. The models were trained on the first 70 points and extrapolated to 200. ANODEs are able to converge to a lower training loss, however they diverge when extrapolating.

As shown in Figure 7, while ANODEs achieve a lower training loss than SONODEs, their test loss is much greater. We conjecture that, in the case of ANODEs, this is a case of overfitting. SONODEs, on the other hand, can better approximate the dynamics, therefore they exhibit better predictive performance. Note that, neither model can learn the VDP oscillator particularly well, which may be attributed to chaotic behaviour of the system at hand.

## D.3 Third Order NODEs on Airplane Vibrations

We test Third Order Neural ODEs (TONODEs) on the Airplane Vibrations task from Appendix D.1. The results are in Figure 8.

We see that TONODEs vastly underperform compared to ANODEs and SONODEs. In each of the 3 repetitions of the experiment, the different initialisation found the best solution to be at zero. Therefore, whilst the loss stays constant, the error remains large. We hypothesise that despite theoretically being able to perform at least as well as SONODEs, TONODEs avoid exponentially growing at any point by exponentially decaying towards zero. It is likely that by rescaling the time to be between 0 and 1, TONODE would approach a more accurate solution.

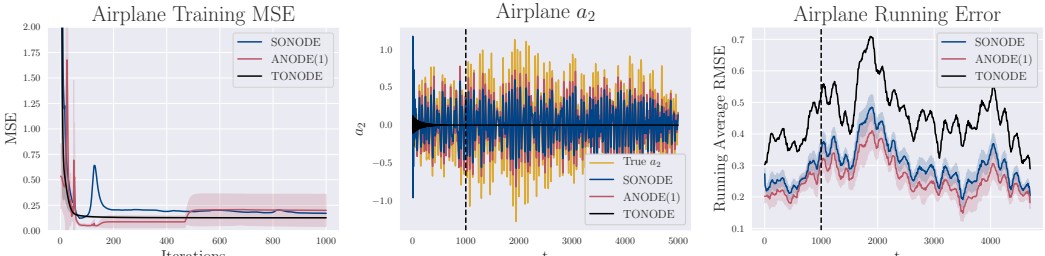

Figure 8: Repeating the Airplane Vibrations task with third order NODEs (TONODEs): training loss curves *(left)*, predicted value *(middle)*, and running error *(right)*. We see that, in this case, TONODEs are not as successful at modelling these dynamics as SONODEs and ANODEs, having a larger error both on the training data and the extrapolation.

## D.4 First Order Dynamics and Interpolation

SONODEs contain NODEs as a subset of models. Consider first order dynamics that is approximated by the NODE

$$\dot{\mathbf{x}} = f^{(v)}(\mathbf{x}, t, \tilde{\theta}_f) \tag{89}$$

Carrying out the full time derivative of Equation (89):

$$\ddot{\mathbf{x}} = \frac{\partial f^{(v)}(\mathbf{x}, t, \tilde{\theta}_f)}{\partial \mathbf{x}^T} \dot{\mathbf{x}} + \frac{\partial f^{(v)}(\mathbf{x}, t, \tilde{\theta}_f)}{\partial t}, \qquad \dot{\mathbf{x}}(t_0) = f^{(v)}(\mathbf{x}(t_0), t_0, \tilde{\theta}_f) \tag{90}$$

Which yields the SONODE equivalent of the learnt dynamics:

$$f^{(a)}(\mathbf{x}, \mathbf{v}, t, \theta_f) = \frac{\partial f^{(v)}(\mathbf{x}, t, \tilde{\theta}_f)}{\partial \mathbf{x}^T} \mathbf{v} + \frac{\partial f^{(v)}(\mathbf{x}, t, \tilde{\theta}_f)}{\partial t}, \qquad g(\mathbf{x}(t_0), \theta_g) = f^{(v)}(\mathbf{x}(t_0), t_0, \tilde{\theta}_f) \tag{91}$$

Additionally, it was shown in Equation (3) that SONODEs are a specific case of ANODEs that learn the initial augmented position. Therefore, anything that NODEs can learn, SONODEs should also be able to learn, and anything SONODEs can learn, ANODEs should be able to learn. To demonstrate that SONODEs and ANODEs can also learn first order dynamics, we task them with learning an exponential with no noise, $x(t) = exp(0.1667t)$. All models, as expected, are able to learn the function, as shown in Figure 9.

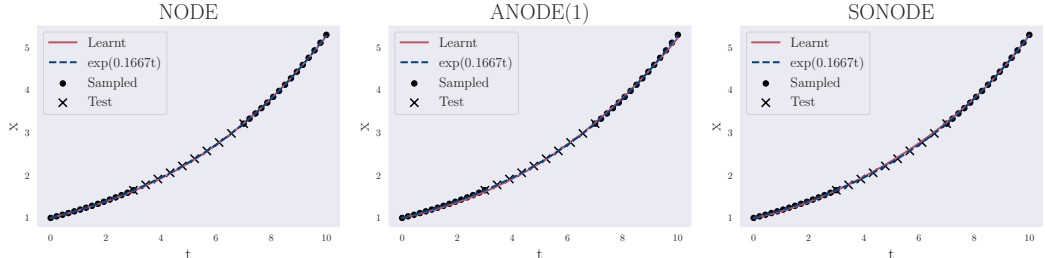

Figure 9: NODE *(left)*, ANODE(1) *(middle)* and SONODE *(right)* learning an exponential (simple first order dynamics) and interpolating between two observation sections. As expected, all models are able to learn the function.

## D.5 Performance on MNIST

NODEs, SONODEs and ANODEs were tested on MNIST [9] to investigate their ability on classification tasks. The networks used convolutional layers, which in the case of SONODEs were used for both the acceleration and the initial velocity. ANODEs were augmented with one additional channel as is suggested by Dupont et al. [4]. The models used a training batch size of 128 and test batch size of 1000, as well as group normalisation. SGD optimiser was used with a learning rate of 0.1 and

Table 1: Results for the MNIST experiments at convergence. SONODE converges to a higher test accuracy than NODEs with a lower NFE. ANODEs converge to the same higher test accuracy with a higher NFE, but with a lower parameter count than SONODEs.

| Model | Test Accuracy | NFE |
|---|---|---|
| NODE | $0.9961 \pm 0.0004$ | $26.2 \pm 0.0$ |
| SONODE | $\mathbf{0.9963 \pm 0.0001}$ | $\mathbf{20.1 \pm 0.0}$ |
| ANODE | $\mathbf{0.9963 \pm 0.0001}$ | $32.2 \pm 0.0$ |

momentum 0.9. The cross-entropy loss was used. The experiment was repeated 3 times with random initialisations to obtain a mean and standard deviation. The results are given in table 1 and Figure 10.

In terms of test accuracy, SONODEs and ANODEs perform marginally better than NODEs. ANODEs can achieve the same accuracy with fewer parameters than SONODEs because the dynamics are not limited to second order and it is only the final state that is of concern in classification. However, SONODEs are able to achieve the same accuracy with a lower number of function evaluations (NFE). NFE denotes how many function evaluations are made by the ODE solver, and represents the complexity of the learnt solution. It is a continuous analogue of the depth of a discrete layered network. In the case of NODEs and ANODEs, the NFE gradually increases meaning that the complexity of the flow also increases. However, in the case of SONODEs, the NFE stays constant, suggesting that the initial velocity was associated with larger gradients (otherwise we would expect NFE to increase for SONODEs with training).

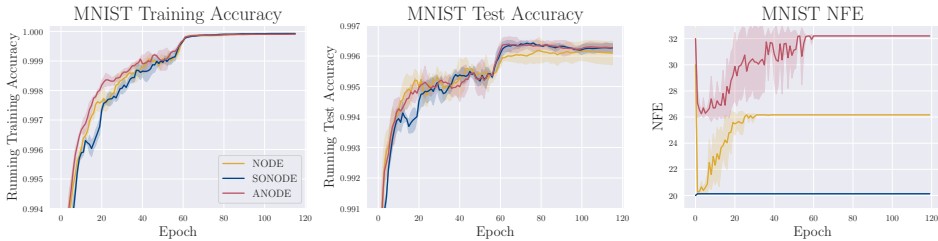

Figure 10: Comparing the performance of SONODEs and NODEs on the MNIST dataset: train accuracy *(left)*, test accuracy *(middle)*, NFE *(right)*. SONODEs converge to the same training accuracy and a higher test accuracy with a lower NFE than NODEs. NODEs had 208266 parameters, SONODEs had 283658 and ANODEs had 210626. Additional parameters were associated with the initial velocity, or the augmented channel.

## E    ANODEs learning 2nd Order

Here we demonstrate the different ways that ANODEs and SONODEs learn second order dynamics, and how this affects their solutions.

**Proposition E.1.** *The general form ANODEs learn second order behaviour is given by:*

$$\begin{bmatrix} \dot{\mathbf{x}} \\ \dot{\mathbf{a}} \end{bmatrix} = \begin{bmatrix} F(\mathbf{x}, \mathbf{a}, t, \theta_F) \\ G(\mathbf{x}, \mathbf{a}, t, \theta_G) \end{bmatrix}, \qquad G = \left( \frac{\partial F}{\partial \mathbf{a}^T} \right)_{left}^{-1} \left( f^{(a)} - \frac{\partial F}{\partial \mathbf{x}^T} F - \frac{\partial F}{\partial t} \right) \tag{92}$$

*Proof.* Let $\mathbf{z}(t)$ be the state vector $[\mathbf{x}(t), \mathbf{a}(t)]$. The time derivatives can be written as

$$\begin{bmatrix} \dot{\mathbf{x}}(t) \\ \dot{\mathbf{a}}(t) \end{bmatrix} = \begin{bmatrix} F(\mathbf{x}, \mathbf{a}, t, \theta_F) \\ G(\mathbf{x}, \mathbf{a}, t, \theta_G) \end{bmatrix} \tag{93}$$

Let $\mathbf{x}(t)$ follow the second order ODE, $\ddot{\mathbf{x}} = \dot{F} = f^{(a)}(\mathbf{x}, \dot{\mathbf{x}}, t, \theta_f)$. Differentiating $F$ with respect to time

$$\dot{F} = \frac{\partial F}{\partial \mathbf{x}^T} \dot{\mathbf{x}} + \frac{\partial F}{\partial \mathbf{a}^T} \dot{\mathbf{a}} + \frac{\partial F}{\partial t} = f^{(a)}(\mathbf{x}, \dot{\mathbf{x}}, t, \theta_f) \tag{94}$$

Using $\dot{\mathbf{x}} = F$ and $\dot{\mathbf{a}} = G$

$$f^{(a)}(\mathbf{x}, F, t, \theta_f) = \frac{\partial F}{\partial \mathbf{x}^T} F + \frac{\partial F}{\partial \mathbf{a}^T} G + \frac{\partial F}{\partial t} \tag{95}$$

Rearranging for G

$$G(\mathbf{x}, \mathbf{a}, t, \theta_G) = \left(\frac{\partial F}{\partial \mathbf{a}^T}\right)^{-1}_{\text{left}} \left(f^{(a)}(\mathbf{x}, F, t, \theta_f) - \frac{\partial F}{\partial \mathbf{x}^T} F - \frac{\partial F}{\partial t}\right) \tag{96}$$

$\square$

In order for the solution of $G$ to exist, the matrix $\dfrac{\partial F}{\partial \mathbf{a}^T}$ must be invertible. Either the dimension of $\mathbf{a}$ matches $F$, $\mathbf{x}$ and $f^{(a)}$, so that $\dfrac{\partial F}{\partial \mathbf{a}^T}$ is square, or $\dfrac{\partial F}{\partial \mathbf{a}^T}$ has a left inverse. Crucially, $F$ must have explicit $\mathbf{a}$ dependence, or the inverse does not exist. Intuitively, in order for real space to couple to augmented space, there must be explicit dependence. Using the left inverse, we do not require $\dfrac{\partial F}{\partial \mathbf{a}^T}$ to be square, so $\mathbf{a}$ does not necessarily need to be the same dimensionality as $F$ and $\mathbf{x}$. ANODEs can learn second order dynamics with fewer dimensions than SONODEs.

Using the equation for $G(\mathbf{x}, \mathbf{a}, t, \theta_G)$, there is a gauge symmetry in the system, which proves proposition E.2.

**Proposition E.2.** *ANODEs can learn an infinity of (non-trivial) functional forms to learn the true dynamics of a second order ODE in real space.*

*Proof.* Assume a solution for $F(\mathbf{x}, \mathbf{a}, t, \theta_F)$ and $G(\mathbf{x}, \mathbf{a}, t, \theta_G)$ has been found such that, $\dot{F} = f^{(a)}$ and $F(\mathbf{x}_0, \mathbf{a}_0, t_0, \theta_F) = \dot{\mathbf{x}}_0$. If an arbitrary function of $\mathbf{x}$, $\phi(\mathbf{x})$, is added to $F$, where $\phi(\mathbf{x}_0) = 0$

$$\tilde{F}(\mathbf{x}, \mathbf{a}, t, \theta_F) = F(\mathbf{x}, \mathbf{a}, t, \theta_F) + \phi(\mathbf{x}) \tag{97}$$

The initial velocity is still the same. The dynamics are preserved if there is a corresponding change in $G$

$$\tilde{G}(\mathbf{x}, \mathbf{a}, t, \theta_G) = \left(\frac{\partial(F + \phi)}{\partial \mathbf{a}^T}\right)^{-1} \left(f^{(a)}(\mathbf{x}, F + \phi, t, \theta_f) - \frac{\partial(F + \phi)}{\partial \mathbf{x}^T}(F + \phi) - \frac{\partial(F + \phi)}{\partial t}\right) \tag{98}$$

The proof can end here, however this can be simplified. $\phi(\mathbf{x})$ has no explicit $\mathbf{a}$ or $t$ dependence, so this equation simplifies to

$$\tilde{G} = \left(\frac{\partial F}{\partial \mathbf{a}^T}\right)^{-1} \left(f^{(a)}(\mathbf{x}, F + \phi, t, \theta_f) - \frac{\partial F}{\partial \mathbf{x}^T} F - \frac{\partial F}{\partial t} - \frac{\partial F}{\partial \mathbf{x}^T} \phi - \frac{\partial \phi}{\partial \mathbf{x}^T} F - \frac{\partial \phi}{\partial \mathbf{x}^T} \phi\right) \tag{99}$$

The term $f^{(a)}(\mathbf{x}, F + \phi, t, \theta_f)$ can be Taylor expanded (assuming convergence)

$$f^{(a)}(\mathbf{x}, F + \phi, t, \theta_f) = f^{(a)}(\mathbf{x}, F, t, \theta_f) + \sum_{n=1}^{\infty} \left(\frac{\partial^n f^{(a)}(\mathbf{x}, \dot{\mathbf{x}}, t, \theta_f)}{\partial \dot{\mathbf{x}}^{Tn}}\bigg|_{\dot{\mathbf{x}}=F} \frac{\phi^n}{n!}\right) \tag{100}$$

Which gives the corresponding change in $G$

$$\tilde{G} = G(\mathbf{x}, \mathbf{a}, t, \theta_G) + \left(\frac{\partial F}{\partial \mathbf{a}^T}\right)^{-1} \left(\sum_{n=1}^{\infty} \left(\frac{\partial^n f^{(a)}}{\partial \dot{\mathbf{x}}^{Tn}}\bigg|_{\dot{\mathbf{x}}=F} \frac{\phi^n}{n!}\right) - \frac{\partial F}{\partial \mathbf{x}^T} \phi - \frac{\partial \phi}{\partial \mathbf{x}^T} F - \frac{\partial \phi}{\partial \mathbf{x}^T} \phi\right) \tag{101}$$

$\square$

This demonstrates that there are infinite functional forms that ANODEs can learn. This only considered perturbing functions $\phi(\mathbf{x})$. More complex functions can be added that have $\mathbf{a}$ or $t$ dependence, which lead to a more complex change in $G$. By contrast, we now show SONODEs have a unique functional form.

**Proposition E.3.** *SONODEs learn to approximate a unique functional form to learn the true dynamics of a second order ODE in real space.*

*Proof.* Consider a dynamical system

$$\frac{d^2\mathbf{x}}{dt^2} = f(\mathbf{x}, \mathbf{v}, t), \qquad \mathbf{x}(t_0) = \mathbf{x}_0, \qquad \mathbf{v}(t_0) = \mathbf{v}_0 \qquad (102)$$

For these problems we let the loss only depend on the position, if it depends on position and velocity there would be more restrictions. So if it is true when loss only depends on the position, it is also true when it depends on both position and velocity.

Assume that there is another system, that has the same position as a function of time

$$\frac{d^2\tilde{\mathbf{x}}}{dt^2} = \tilde{f}(\tilde{\mathbf{x}}, \tilde{\mathbf{v}}, t), \qquad \tilde{\mathbf{x}}(t_0) = \tilde{\mathbf{x}}_0, \qquad \tilde{\mathbf{x}}(t_0) = \tilde{\mathbf{v}}_0 \qquad (103)$$

Where $f(\mathbf{x}, \mathbf{v}, t) \neq \tilde{f}(\tilde{\mathbf{x}}, \tilde{\mathbf{v}}, t)$. Because the initial conditions are given, the position and velocity are defined at all times, and therefore position, velocity and acceleration can all be written as explicit functions of time. $\mathbf{x} \equiv \mathbf{x}(t)$, $\mathbf{v} \equiv \mathbf{v}(t)$. This allows for the acceleration to be written as a function of $t$ only, $f(\mathbf{x}, \mathbf{v}, t) = f_\tau(t)$ for all $t$. The same applies for the second system, $\tilde{\mathbf{x}} \equiv \tilde{\mathbf{x}}(t)$, $\tilde{\mathbf{v}} \equiv \tilde{\mathbf{v}}(t)$ and $\tilde{f}(\tilde{\mathbf{x}}, \tilde{\mathbf{v}}, t) = \tilde{f}_\tau(t)$.

For all $t$, $\mathbf{x}(t) = \tilde{\mathbf{x}}(t)$, therefore, for any time increment, $\delta t$, $\mathbf{x}(t + \delta t) = \tilde{\mathbf{x}}(t + \delta t)$. Taking the full time derivative of $\mathbf{x}$ and $\tilde{\mathbf{x}}(t)$

$$\frac{d\mathbf{x}(t)}{dt} = \mathbf{v}(t) = \lim_{\delta t \to 0} \frac{\mathbf{x}(t + \delta t) - \mathbf{x}(t)}{\delta t} \qquad (104)$$

$$\frac{d\tilde{\mathbf{x}}(t)}{dt} = \tilde{\mathbf{v}}(t) = \lim_{\delta t \to 0} \frac{\tilde{\mathbf{x}}(t + \delta t) - \tilde{\mathbf{x}}(t)}{\delta t} \qquad (105)$$

Using these two equations and the fact that $\mathbf{x}(t) = \tilde{\mathbf{x}}(t)$, it is inferred that $\mathbf{v}(t) = \tilde{\mathbf{v}}(t)$ for all $t$. Taking the full time derivative of $\mathbf{v}(t)$ and $\tilde{\mathbf{v}}(t)$

$$\frac{d\mathbf{v}(t)}{dt} = f_\tau(t) = \lim_{\delta t \to 0} \frac{\mathbf{v}(t + \delta t) - \mathbf{v}(t)}{\delta t} \qquad (106)$$

$$\frac{d\tilde{\mathbf{v}}(t)}{dt} = \tilde{f}_\tau(t) = \lim_{\delta t \to 0} \frac{\tilde{\mathbf{v}}(t + \delta t) - \tilde{\mathbf{v}}(t)}{\delta t} \qquad (107)$$

Using these two equation and the fact that $\mathbf{v}(t) = \tilde{\mathbf{v}}(t)$ for all $t$, it is also inferred that $f_\tau(t) = \tilde{f}_\tau(t)$ for all $t$.

Using these three facts, $\mathbf{x}(t) = \tilde{\mathbf{x}}(t)$, $\mathbf{v}(t) = \tilde{\mathbf{v}}(t)$ and $f_\tau(t) = \tilde{f}_\tau(t)$. It must also be true that $f(\mathbf{x}(t), \mathbf{v}(t), t) = \tilde{f}(\tilde{\mathbf{x}}(t), \tilde{\mathbf{v}}(t), t) \rightarrow f(\mathbf{x}, \mathbf{v}, t) = \tilde{f}(\mathbf{x}, \mathbf{v}, t)$. Therefore the assumption that $f(\mathbf{x}, \mathbf{v}, t) \neq \tilde{f}(\tilde{\mathbf{x}}, \tilde{\mathbf{v}}, t)$ is incorrect, there can only be one functional form for $f(\mathbf{x}, \mathbf{v}, t)$.

Additionally, using $\mathbf{v}(t) = \tilde{\mathbf{v}}(t)$ for all $t$, the initial velocities must also be the same.

$\square$

