# OpenReview forum: "On Second Order Behaviour in Augmented Neural ODEs: A Short Summary"
_NeurIPS.cc/2021/Workshop/DLDE — DLDE Workshop -- NeurIPS 2021 Poster_

### Official Review · Reviewer_nx8j · 2021-09-29
**This work improves on an existing methodology by shifting a tradeoff. The paper is well written and explores very promising avenues**

**Confidence:** 3

**Review:**

This work proposes using Second Order NODEs (SONODEs), which are a modification of the existing Augmented Neural ODEs. The applicability of SONODEs is well explained and the overall structure of the work is commendable. However, I do have two questions:

1) So ANODEs just append extra dimensions to account for attributes which help in, say a classification task and in SONODEs you restrict yourselves to just second order dynamics. Isn't this restriction a hurdle? Shouldn't there be a way to find an optimal number of dimension augmentations?

2) Is there a relation between ANODEs/SONODEs and kernelization? Both seem to manipulate the number of dimensions you are working in

Overall, this paper is a great step ahead in the field of neural ODEs and the authors have conveyed their points across efficiently.

**Score:**

4: Very good paper

---

### Official Review · Reviewer_k3Em · 2021-10-01
**Review for "On Second Order Behaviour in Augmented Neural ODEs "**

**Confidence:** 2

**Review:**

In this paper the authors attempt to showed that Second Order Neural ODEs are a special case of Augmented
Neural ODEs with restricted dynamics. The authors show that Augmented Neural ODEs  are more efficient and learn second order dynamics using fewer dimensions compared to Second Order Neural ODEs. On the other hand Second Order Neural ODEs dimensions are lsess entangled compare to Augmented Neural ODEs making them more interpretable.


**Score:**

3: Good paper

---

### Official Review · Reviewer_ewqo · 2021-10-14
**Modeling second order behavior in neural ODEs**

**Confidence:** 2

**Review:**

This paper introduces an augmented ODE architecture to address second-order behavior characteristics in dynamical systems.  The motivation for the work is clear and the problem setup is well framed.  While their formulation outperforms ANODEs, the overall implications of the work are somewhat incremental.  There's also a problem with interpretability since they lead to more entanglement of the dimensions.  For many systems, this property would prevent the application of this approach.  I believe the work is sound and it's interesting that the architecture is able to model beyond what existing techniques can.  However, it's hard to measure what the implications are and it seems that the contribution is at best, incremental.

**Score:**

2: Borderline paper

---

### Decision · Program_Chairs · 2021-10-16

**Decision:**

Accept (Poster)

**Comment:**

Reviewers have recommended acceptance.